# Simulating a chemically fueled molecular motor with nonequilibrium molecular dynamics

Alex Albaugh 🔗 [1] & Todd R. Gingrich 🔗 [1✉]

Most computer simulations of molecular dynamics take place under equilibrium conditions—in a closed, isolated system, or perhaps one held at constant temperature or pressure. Sometimes, extra tensions, shears, or temperature gradients are introduced to those simulations to probe one type of nonequilibrium response to external forces. Catalysts and molecular motors, however, function based on the nonequilibrium dynamics induced by a chemical reaction's thermodynamic driving force. In this scenario, simulations require chemostats capable of preserving the chemical concentrations of the nonequilibrium steady state. We develop such a dynamic scheme and use it to observe cycles of a particle-based classical model of a catenane-like molecular motor. Molecular motors are frequently modeled with detailed-balance-breaking Markov models, and we explicitly construct such a picture by coarse graining the microscopic dynamics of our simulations in order to extract rates. This work identifies inter-particle interactions that tune those rates to create a functional motor, thereby yielding a computational playground to investigate the interplay between directional bias, current generation, and coupling strength in molecular information ratchets.

[1] Department of Chemistry, Northwestern University, 2145 Sheridan Road, Evanston, IL 60208, USA. ✉email: todd.gingrich@northwestern.edu

Molecular motors are ubiquitous in biology. Proteins like kinesin[1] and myosin[2] transduce free energy, hydrolyzing adenosine triphosphate (ATP) to power mechanical work[3–6]. These motors operate by coupling ATP hydrolysis to linear motion, carrying cellular cargoes along microtubule and actin tracks, respectively. Those natural motors have also been engineered to modify their performance. Mutated kinesin can process further along microtubule tracks than wild type[7,8] or rapidly cease activity in response to small molecules[9]. Myosin can be engineered to move along an actin track in the opposite direction of the wild type motor[10,11]. Despite those successes modifying existing motors, it remains challenging to design molecular interactions to build similar machines from the ground up.

Chemists have sought to build those machines using the principles of the biological motors but with different synthetic building blocks[12–14]. Like the biological inspiration, the synthetic machines should rectify thermal fluctuations into directed motion by harvesting free energy from chemical fuel, a goal first realized by the artificial motor of Wilson et al.[15]. One challenge in designing these machines is that the mechanism is typically considered in terms of the kinetics of elementary steps while the design is more naturally thought of in terms of the strength of interactions between molecular components. Connecting those interactions to the ultimate dynamical function is particularly challenging because microscopic motors operate in a noisy regime characterized by stochastic fluctuations[16,17].

In equilibrium situations, computer simulations have proven to be particularly useful at bridging that connection between molecular design and dynamics, particularly in the presence of noise[18]. The nonequilibrium dynamics of molecular motors, however, preclude straightforward application of equilibrium simulation methods. Equilibrium dynamics moves in forward and reverse directions with equal probability, so a directional motor requires nonequilibrium conditions powered by a chemical fuel[3,6,19,20]. To capture the nonequilibrium behavior in simulations, a number of different strategies have been employed. One approach aims to describe different equilibria of a motor, e.g., one with a fuel bound and one with the fuel unbound. The nonequilibrium dynamics is induced by externally imposing time-dependent swaps between these energy surfaces[21–27]. A complementary body of work breaks the time-reversal symmetry of equilibrium dynamics by imposing forces or torques on the motor[28–33]. Both approaches can obscure how the chemistry couples to the mechanical motion, and that mechanochemical coupling is central to a motor's function[6,34]. To explicitly capture that coupling, it is necessary to continually resupply fuel and extract waste from a simulation so as to sustain a nonequilibrium steady state (NESS), a strategy implemented with a minimal kinesin-like walker model[35] and with Janus particles and sphere-dimers motor models that move along self-induced concentration gradients (diffusiophoresis)[36–39].

Here we present a model motor and fuel with sufficiently simple pair potentials that the steady-state dynamics can be directly simulated, with a nonequilibrium environment maintained by external baths. Our motor is essentially that of Wilson et al.[15], where a reaction biases the relative motion of two interlocked rings in a preferred direction. We show how simulations can be used to quantify motor performance and tradeoffs. Armed with the explicit particle-based model, we analyze the resulting currents using a nonequilibrium Markov state framework, with which we aim to more directly connect the stochastic thermodynamic analysis of motors[40–42] with particle-based simulations. The model and methods we report serve as a test-bed for exploring how inter-particle interactions affect the operation of a molecular motor.

## Results

**Fueling a nonequilibrium steady state with a classical fuel model.** Consider the dynamics of a motor protein in the presence of ATP, adenosine diphosphate (ADP), and inorganic phosphate (P). In an ideal solution, the reversible chemical reaction ATP ⇌ ADP + P will relax into an equilibrium with equilibrium constant

$$K = \frac{[\text{ADP}][\text{P}]}{[\text{ATP}]} = e^{-\beta(\mu^0_{\text{ADP}} + \mu^0_{\text{P}} - \mu^0_{\text{ATP}})}, \quad (1)$$

where $\mu^0_{\text{ADP}}$, $\mu^0_{\text{P}}$, and $\mu^0_{\text{ATP}}$ are standard-state chemical potentials and $\beta$ is the inverse temperature in units of Boltzmann's constant $k_{\text{B}}$. At chemical equilibrium, the motion of the motor must obey detailed balance, precluding the protein from exhibiting net motion. The situation is altered if external means prevent the chemical reaction from reaching equilibrium, for example, if ATP is fed into the system while ADP and P are extracted. Provided the reaction of ATP is suitably coupled to the protein's motion, the fuel's free energy gradient pushes the motor into a NESS with net directed motion, giving rise to currents. In so-called tightly coupled motors, each reaction event correlates with a configurational change of the motor. For example, when F$_1$-ATP synthase generates work from ATP[43], each catalyzed ATP hydrolysis corresponds almost one-to-one with a 120° rotation of a rotor[44]. Other motors are loosely coupled, with motor motion only weakly correlated with fuel consumption[45].

That mechanochemical coupling can be realized in a strictly classical model, provided the model exhibits a reversible transformation between fuel and waste and that a continuous influx of fuel and outflow of waste prevents relaxation to equilibrium. It is furthermore necessary that the model fuel exhibit metastability, so that interconversion between fuel and waste is slower than fuel injection and waste removal. We constructed the classical fuel out of tetrahedral clusters of volume-excluding particles, as shown in Fig. 1. Four such particles, colored blue, are bonded together to form a tetrahedral shell. A single unbound volume-excluding particle, colored red, can be kinetically trapped inside the tetrahedron. A filled tetrahedral cluster (FTC) does not retain its red central particle (C) indefinitely. Rather, a rare thermal fluctuation inevitably allows the tetrahedral cluster to contort enough for the kinetically trapped C to escape, leaving behind an empty tetrahedral cluster (ETC). Consistent with microscopic reversibility, the reverse process is also possible. At equilibrium, the flux from FTC → ETC + C would balance the reverse flux of ETC + C → FTC. Since the ETC + C state is both entropically and energetically favorable, equilibrium would strongly favor the empty tetrahedra. An initial concentration of FTC fuel would quickly deplete to its near-zero equilibrium concentration if not for grand canonical Monte Carlo (GCMC) chemostats, which provide a mechanism to hold the chemical potentials for the three different species at unequilibrated values. Consequently, within a simulation cell, the FTC, ETC, and C species are stochastically injected and removed so as to maintain a NESS in which the FTC → ETC + C reaction is typical. The reverse reaction, though possible, is practically unobserved. Because the statistical consequences of nonequilibrium driving forces are present even in strictly classical dynamics, it is not necessary to confront the quantum-mechanical complexities presented by chemical bond breaking. Rather, the classical model suffices as a practical way to address fundamental questions about the impact of pairwise interactions on dynamical function.

**A classical motor model.** We aimed to engineer a model motor capable of harvesting the free energy of a NESS with a high concentration of FTC and low concentrations of ETC and C,

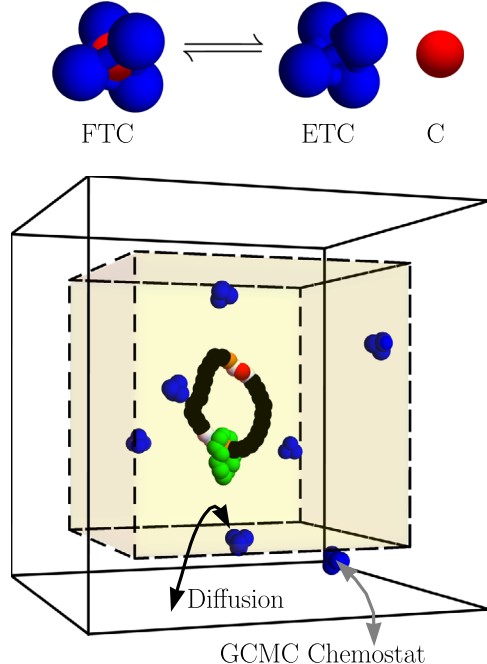

**Fig. 1 Molecular motor model with nonequilibrium simulation setup.** Top: a tetrahedral cluster formed from two types of particles can transition between filled tetrahedral cluster (FTC) and empty tetrahedral cluster (ETC) plus central particle (C) states while executing Langevin dynamics at a fixed temperature. Chemical potentials for each species ($\mu_{FTC}$, $\mu_{ETC}$, and $\mu_C$) are regulated with grand canonical Monte Carlo chemostats to drive the reaction away from equilibrium. Bottom: a simulation cell containing a model motor driven at the nonequilibrium steady state fuel concentrations. The motor is constrained to occupy the inner box (shaded yellow) through a Lennard–Jones wall potential while the GCMC chemostats insert and remove FTC, ETC, and C only from the space between the inner and outer boxes (white background). FTC, ETC, and C do not experience the Lennard–Jones wall potential and freely diffuse between the inner and outer boxes. The motor consists of a small shuttling ring (green particles) that diffuses around a larger ring composed of two shuttling-ring–binding sites (orange particles), each adjacent to a three-particle catalytic site (white particles). The remainder of the larger ring is made of inert particles (black) that only have mildly repulsive interactions with other particles. The catalytic sites speed up the FTC → ETC + C decomposition due to attractions between white catalytic particles and the blue and red particles of FTC. Upon their escape, the red C particles can linger around the catalytic sites, blocking the shuttling ring and ultimately gating diffusion to generate net directed current.

motivated by the first synthetic, autonomous, chemically fueled molecular motor of Wilson et al.[15]. Their motor is a catenane consisting of two interlocked rings. The smaller of the two rings, a benzylic amide, shuttles around a track formed by the larger ring. On that track, Wilson et al. engineered two fumaramide binding sites as well as two adjacent hydroxyl groups that catalyze the decomposition of a bulky fuel (9-fluorenylmethoxycarbonyl chloride) into waste products ($CO_2$ and dibenzofulvene). The relative positioning of binding and catalytic sites breaks symmetry such that fuel reaction induces directed motion, the kinetics of which have been expressed elegantly in terms of an information ratchet[46,47], where directed motion arises from the gating of natural thermal diffusion in a preferred direction[3,48–50]. That mechanism relies on steric considerations; the fuel reacts more slowly at a catalytic site when the shuttling ring is near enough to block access to the catalytic site. The same sort of mechanism underlies our coarse-grained, classical design. The kinetics of

catalyzed fuel reactions must be sensitive to the proximity of the shuttling ring.

In our model, that need is satisfied by introducing inter-molecular interactions between the shuttling ring and the components of the model fuel. As described briefly in Fig. 1 and more thoroughly in Methods, we construct a motor from two interlocking rings of particles. The smaller green ring has attractive interactions with orange binding sites on the larger ring. The particles of FTC, ETC, and C molecules have interactions that encourage the FTC → ETC + C reaction at the white catalytic sites. Following the reaction, the C particle remains at the catalytic site as a blocking group, which the shuttling ring cannot diffuse past. Proximity of the shuttling ring to a catalytic site decreases the rate of catalysis relative to the distal catalytic site. This imbalance of rates, along with the nonequilibrium replenishment of FTC and removal of ETC and C, yields net directed motion when the pair potentials are suitably tuned, a point we return to in a more detailed discussion of the mechanism.

**Dynamics.** The dynamics of the fueled motor were evolved in time by mixing the Langevin dynamics of the particles with GCMC chemostats that maintained the NESS. The Langevin equations of motion for each particle $i$ are

$$\dot{\mathbf{r}}_i = \frac{\mathbf{p}_i}{m_i}$$
$$\dot{\mathbf{p}}_i(t) = -\nabla U(\mathbf{r}_i(t)) - \frac{\gamma}{m_i}\mathbf{p}_i(t) + \boldsymbol{\xi}_i(t), \tag{2}$$

where $\gamma$ is the friction coefficient, $\mathbf{p}_i$ is the momentum of particle $i$, $\mathbf{r}_i$ is the position of particle $i$, $m_i$ is the mass of that particle, $U$ is the potential energy, and $\boldsymbol{\xi}_i$ is white noise with $\langle\boldsymbol{\xi}_i\rangle = \mathbf{0}$ and $\langle\boldsymbol{\xi}_i(t)\boldsymbol{\xi}_i(t')\rangle = 2\gamma k_B T\delta(t-t')$ at temperature $T$. All model parameters are reported in non-dimensional units as described in Methods.

The simulation box consists of two concentric cubes with an inner cube and an outer cube, shown in Fig. 1. GCMC moves occur between the inner and outer boxes and serve to insert and remove FTC, ETC, and C from the system. The motor itself (the two interlocked rings) is confined to the inner box with a wall potential, but the wall potential is not applied to the FTC, ETC, or C molecules, which freely diffuse between the two boxes and can cross the periodic boundaries of the outer box. Since GCMC insertions and deletions occur in the space between the inner and outer box and the motor is confined to the inner box, the motor will not be directly affected by the GCMC moves. However, the motor does feel the indirect effect of the nonequilibrium concentrations since the timescale for diffusion is fast compared to the lifetime of the FTC. After every 100 time steps of Langevin dynamics, a GCMC trial move is chosen uniformly from six options—an insertion or deletion of the three species: FTC, ETC, or C. These moves are conditionally accepted with a Metropolis factor that depends on the set chemical potentials of the three species and their instantaneous concentrations. As described in Methods and the Supplementary Information (SI), the GCMC procedure must account for the internal degrees of freedom of the FTC and ETC clusters[51,52]. Due to those internal degrees of freedom, the GCMC acceptance probabilities directly depend on $\mu' \equiv \mu - A^0$, the applied external chemical potential less the standard-state Helmholtz free energy. The strongly driven regime corresponds to having a high $\mu'_{FTC}$ but a low $\mu'_{ETC}$ and $\mu'_C$. Under those conditions, the typical process starts by inserting FTC into the outer box. This FTC diffuses into the inner box where it interacts with the motor and gets converted into ETC and C.

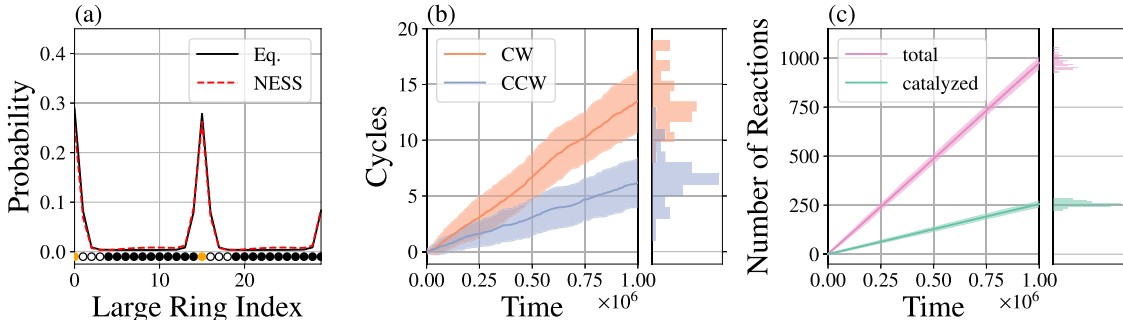

**Fig. 2 Behavior of a motor away from equilibrium.** Motor II was simulated with $\mu'_{FTC} = 0.5$, $\mu'_{ETC} = -10$, and $\mu'_C = -10$ corresponding to an average of about 8.5 FTC molecules, 0.1 ETC molecules, and 0.9 C particles in the simulation box. **a** The distribution for the shuttling ring position, measured as the index of large ring particle nearest to the shuttling ring center of mass, for NESS and equilibrium (Eq.) conditions in the absence of FTC, ETC, and C. An unfolded schematic of the large ring is superimposed over the particle indices to visualize the particle types at each location around the large ring. Orange particles are binding sites, white particles are catalytic sites, and black particles are inert. **b** In the NESS conditions, the shuttling ring executes more clockwise (CW, orange) than counterclockwise (CCW, blue) cycles around the large ring when measured in the frame of the large ring. **c** Decomposition reactions FTC → ETC + C include those catalyzed by the motor (green) and spontaneous reactions in the bulk, which combine to give the total (pink). Decompositions occurring within 2 distance units of a white catalytic particle are classified as having been catalyzed by the motor. Means and standard deviations are generated from 50 independent simulations for each motor.

These waste products then diffuse back into the outer box where they are removed by the GCMC chemostats.

**Bias, current, and coupling efficiency.** The motor and fuel models are characterized by numerous parameters controlling the form and strength of pairwise interactions. We first discovered parameters for the fuel that resulted in the desired metastability of the FTC state. Subsequently, we scanned parameter spaces to identify the interactions between motor and fuel that would reliably generate current, landing upon two sets of interactions, herein referred to as Motor I and Motor II. These two motors differ only subtly; Motor I features slightly stronger attractions between the shuttling ring and binding and also between the C particles and the catalytic site. The full parameterization of both motors can be found in Appendix D of the SI.

The behavior of Motor II in an underdamped regime ($\gamma = 0.5$) with a moderate driving force is shown in Fig. 2 (see also Supplementary Movie 1). The NESS fuel concentration only slightly alters the distribution of the motor configurations relative to equilibrium with no FTC, ETC, or C present. In both cases, the steady-state location of the shuttling ring concentrates around the binding sites. Despite that similarity between the equilibrium and NESS stationary distributions, the NESS dynamical behavior deviates markedly from equilibrium. In the presence of the NESS driving, the total number of clockwise (CW) and counter-clockwise (CCW) cycles do not balance, corresponding to net current. Figure 2 also reflects two important manners in which the present model motor differs from biological machines like ATP synthase. Firstly, our motor is fairly loosely coupled— Fig. 2b, c shows that a single net cycle requires approximately 35 catalyzed FTC → ETC + C reactions. Secondly, the model fuel is less deeply metastable than ATP. Even in the absence of a motor's catalytic site, FTC can degrade on simulation timescales. As such, Fig. 2c distinguishes between catalyzed decompositions that occur in proximity to the catalytic sites and the total decompositions that could occur elsewhere.

In Fig. 3, we report how adding more fuel increases the CW bias, increases the current, and decreases the coupling. Those three measures of motor performance were calculated by monitoring the number of CW and CCW shuttling ring cycles, $n_{CW}$ and $n_{CCW}$, respectively. If the motor's goal is to generate CW cycles then one measure of accuracy is the CW bias, the fraction

of cycles in the CW direction:

$$\frac{n_{CW}}{n_{CW} + n_{CCW}}. \tag{3}$$

The current, the net cycles per time, is similarly computed from $n_{CW}$ and $n_{CCW}$ as

$$\frac{n_{CW} - n_{CCW}}{t_{obs}}, \tag{4}$$

where $t_{obs} = N_{steps} \Delta t$ is the observed simulation time and $N_{steps}$ is the number of simulation time steps of size $\Delta t$. Finally, the coupling between catalyzed reaction and net CW cycles is

$$\frac{n_{CW} - n_{CCW}}{n_{cat}}, \tag{5}$$

where $n_{cat}$ counts the number of FTC decompositions occurring with center of mass within 2 units of a catalytic particle.

Both motors exhibit similar responses to changes in FTC concentration, illustrating a tradeoff: greater bias comes at the expense of lower current and lower coupling. We anticipated a maximum coupling of 0.5, corresponding to a tightly coupled cycle with one catalyzed reaction at each catalytic site. Neither motor achieves that limit. Rather, they are loosely coupled, with catalyzed reactions probabilistically gating diffusion and inducing no major conformational changes in the motor itself. Though the coupling efficiency of these motors is about one order of magnitude below the maximum, we find it encouraging that such a crudely designed toy model can nevertheless convert roughly 1/10 of the catalyzed reactions into directed current.

Since we have described simulations in the underdamped regime ($\gamma = 0.5$), it is natural to wonder if the motor's current is dependent on inertia. Figure 4 shows that the current generation indeed persists in an overdamped regime ($\gamma = 10$) more reflective of the viscous low Reynolds number environment experienced by in vivo biological motors. While increased damping reduces the current by an order of magnitude, it also causes the CW bias of both motors to increase, with Motor I approaching 100%.

**An eight-state rate model.** To rationalize the dynamics of the explicit NESS simulations, it is productive to analyze the rates for transitioning between discrete coarse-grained states. Inspired by a simple six-state Markov model[46,47] that captures the mechanism of the Wilson et al. motor[15], we harvested our simulation data to

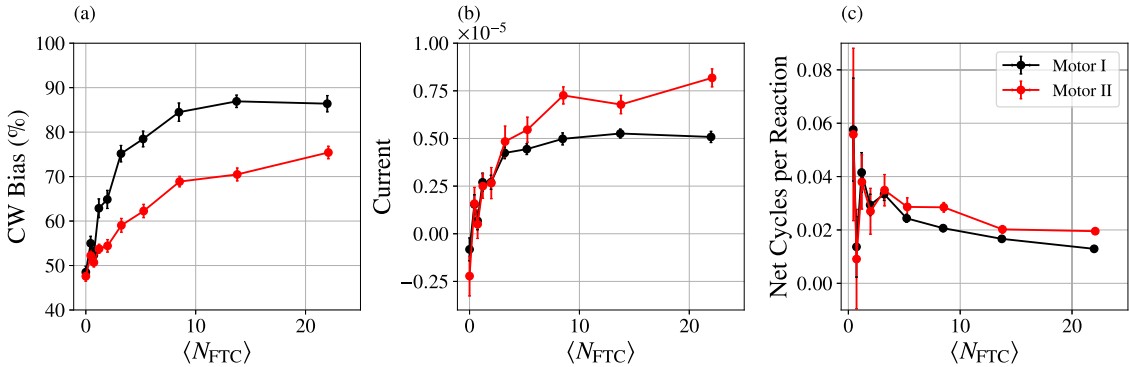

**Fig. 3 Response of the motors to fuel concentration.** By adjusting $\mu'_{\text{FTC}}$ from −1 to 1 in increments of 0.25 and holding fixed $\mu'_{\text{ETC}} = \mu'_{\text{C}} = -10$, the typical number of fuel molecules $\langle N_{\text{FTC}} \rangle$ was tuned in accordance with Eq. (B10). The clockwise bias (**a**, Eq. (3)), current (**b**, Eq. (4)), and coupling (**c**, Eq. (5)) of the two motors were computed from 50 independent simulations of $t_{\text{obs}} = 1 \times 10^6$ units of time each. Error bars represent the standard error across the 50 simulations.

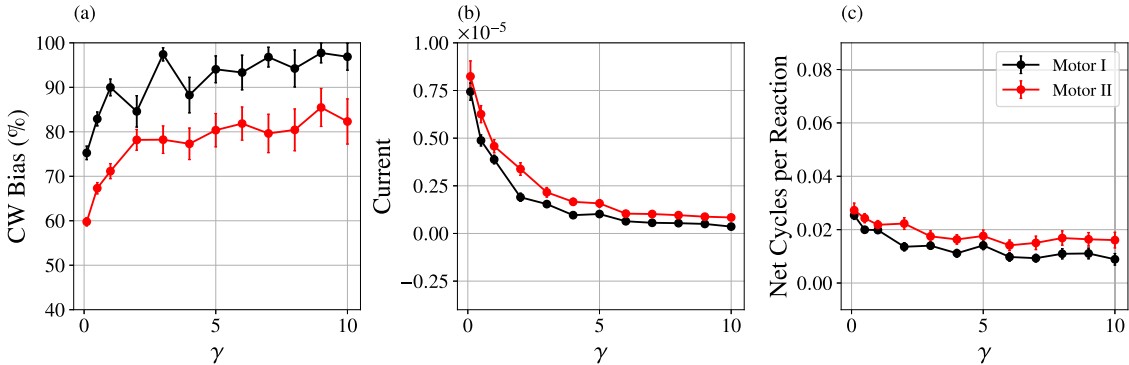

**Fig. 4 Response of the motors to increased damping.** For a fixed NESS with $\mu'_{\text{FTC}} = 0.5$, $\mu'_{\text{ETC}} = \mu'_{\text{C}} = -10$ (corresponding to $\langle N_{\text{FTC}} \rangle = 8.6$, $\langle N_{\text{ETC}} \rangle = 0.1$, and $\langle N_{\text{C}} \rangle = 0.9$), clockwise bias (**a**), current (**b**), and coupling (**c**) were analyzed from simulations with variable friction coefficient $\gamma$. Reported data are the means and standard errors of 50 independent simulations of $t_{\text{obs}} = 1 \times 10^6$ units of time each. As the friction increases, the overall rates all decrease, causing some high-friction simulations to have no cycles in either direction. Those trials were excluded from the clockwise bias statistics, but were still included in the calculation of the current and the net cycles per catalyzed reaction.

collect statistics of the transition times between the eight coarse-grained states depicted in Fig. 5. Those states are determined by three bits of information: (1) which half of the large ring is nearest the shuttling ring center of mass, (2) whether the first catalytic site is blocked, and (3) whether the second catalytic site is blocked, with blockage defined as having at least one free C within 1.2 distance units of a catalytic site's middle particle.

Due to the symmetry of the problem, we focus on seven rates for transitions between these eight states: $k_{\text{attach,close}}$ and $k_{\text{cleave,close}}$ for addition and removal of a blocking group at the catalytic site nearest the shuttling ring, $k_{\text{attach,far}}$ and $k_{\text{cleave,far}}$ for the addition and removal rates from the catalytic site farthest from the shuttling ring, $k_{\text{CW}}$ and $k_{\text{CCW}}$ for CW and CCW rotations of the shuttling ring when one catalytic site has a blocking group, and $k_{\text{sym}}$ for rotations of the shuttling ring when no blocking groups are present. The rates $k_{\text{CW}}$ and $k_{\text{CCW}}$ unambiguously imply a direction of shuttling ring motion, while $k_{\text{sym}}$ results in an even split between CW and CCW. At each NESS simulation time step, the motor's configuration is classified as one of the eight states. If one makes a Markovian assumption, the rate for the transition from coarse-grained state $A$ to state $B$ is

$$k_{AB} = \frac{1}{p_{\text{ss}}(A)} \frac{N_{AB}}{t_{\text{obs}}}. \quad (6)$$

Here $p_{\text{ss}}(A)$ is the steady state probability of being in state $A$ and $N_{AB}$ is the number of transitions from $A$ to $B$ observed in time $t_{\text{obs}}$. To extract the best rate estimate, transitions that are

statistically equivalent by symmetry were combined, e.g., $k_{\text{cleave,far}} = \frac{1}{t_{\text{tobs}}} \frac{N_{21} + N_{34} + N_{56} + N_{87}}{p_{\text{ss}}(2) + p_{\text{ss}}(5) + p_{\text{ss}}(8)}$. Because we simulated a soft system with finite time steps, a transition between two disconnected states of Fig. 5 was very occasionally observed, but we neglected these transitions when constructing the rate model.

To analyze how the interplay between rates generates current, it is productive to decompose the eight-state rate model into four fundamental cycles (FC1–FC4), shown in Fig. 5. Any possible cycle on the graph can be formed by a linear combination of this (non-unique) set of fundamental cycles. Only FC1 gates shuttling ring diffusion into directed motion at both catalytic sites. Traversing FC1 in the CW direction implies that the shuttling ring completes one CW cycle. A CW traversal around FC2 or FC3 similarly corresponds to CW shuttling ring cycling. However, the CW bias is only half that of FC1 because the shuttling ring direction is ambiguous when FC2 and FC3 pass through the unblocked states 4 and 6. The final cycle, FC4, is a futile cycle. Despite burning fuel to traverse FC4, no net cycles of the shuttling ring are generated.

An advantage of the fundamental cycle perspective is that the direction of the steady state currents follows from the ratio of rates around the closed fundamental cycles. For example, fundamental cycles FC2 and FC3 share the ratio

$$R = \frac{k_{\text{attach,far}}}{k_{\text{attach,close}}} \frac{k_{\text{cleave,close}}}{k_{\text{cleave,far}}} \frac{k_{\text{CW}}}{k_{\text{CCW}}}. \quad (7)$$

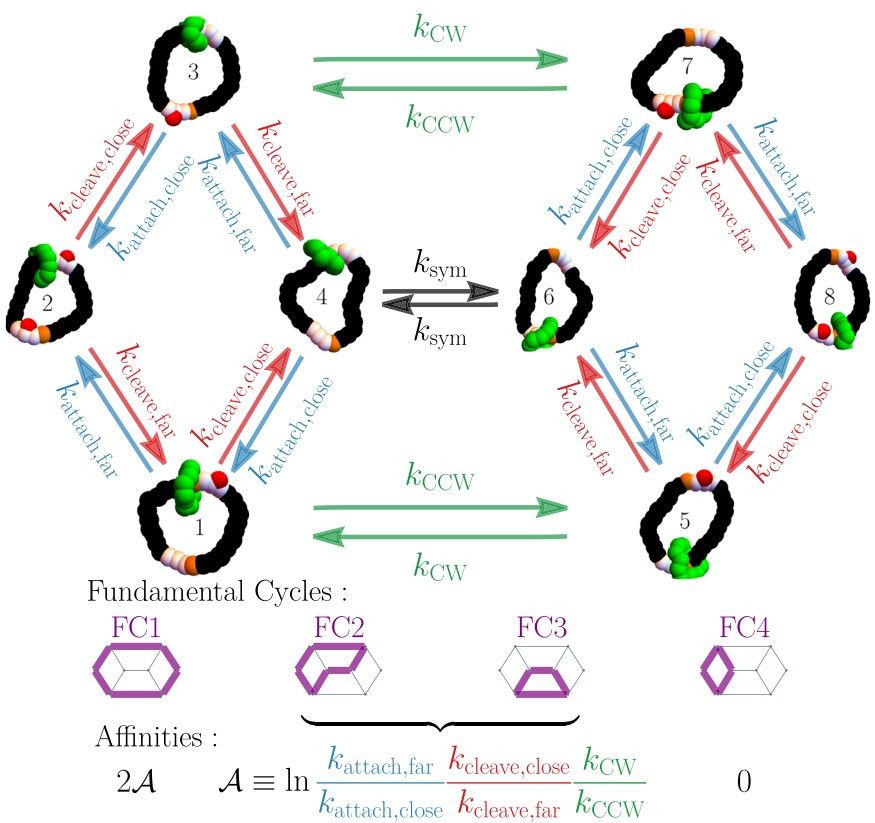

**Fig. 5 A coarse-grained model for understanding motor mechanism.** Motor dynamics can be coarse grained onto an eight-state kinetic model based on the position of the shuttling ring relative to the two binding sites and the presence or absence of at least one C particle blocking group at each catalytic site. Due to symmetries there are seven macroscopic rates in the system, labeled as transitions between certain states. The kinetic model decomposes into four fundamental cycles, shown in purple, with mean steady-state currents oriented in the same direction as the cycle affinities.

We call the logarithm of this ratio the cycle affinity $\mathcal{A} = \log R$, and note that the steady state current around a FC must share the same sign as $\mathcal{A}$[53]. Because all four FCs have a cycle affinity that is a non-negative multiple of $\mathcal{A}$, the steady state current's sign is inherited from the sign of $\mathcal{A}$. Put more succinctly in terms of R, if $R > 1$ the shuttling ring will move CW and if $R < 1$ the shuttling ring will move CCW.

**Clockwise directionality**. We develop our understanding of the motor's CW motion by building off an equilibrium reference, for which $R = 1$ is required by time-reversal symmetry. There are multiple ways to construct an equilibrium reference. For example, we could simulate the motor's equilibrium behavior when $\langle N_{\text{FTC}} \rangle = \langle N_{\text{ETC}} \rangle = \langle N_{\text{C}} \rangle = 0$. With no C particles that equilibrium would confine the motor to states 4 and 6. Instead, we constructed a reference state with non-vanishing $\langle N_{\text{C}} \rangle$ and with $\langle N_{\text{FTC}} \rangle \approx \langle N_{\text{ETC}} \rangle \approx 0$ by setting $\mu'_{\text{C}} = -3$ and $\mu'_{\text{FTC}} = \mu'_{\text{ETC}} = -10$. In this way all eight coarse-grained states and all transitions are observed in the reference (which is essentially equivalent to an equilibrium simulation with a single $\mu'_{\text{C}} = -3$ chemostat). We bias away from this equilibrium by increasing $\mu'_{\text{FTC}}$.

Figure 6a shows that only the two rate constants regulating the blocking group attachment ($k_{\text{attach,close}}$ and $k_{\text{attach,far}}$) respond strongly to the fuel injection. Those attachment rates are functions of the fuel concentration, as one might expect from mass action kinetics when FTC reacts at the catalytic sites to leave behind C as a blocking group. Across the range of FTC concentration, the other five rates behave effectively the same as in the $\langle N_{\text{FTC}} \rangle = 0$ equilibrium reference state. To emphasize that the attachment rates are functions of fuel concentration, we adopt the notation $k_{\text{attach},*}(\langle N_{\text{FTC}} \rangle)$. No argument is needed for

the other rates because those rates are effectively independent of FTC concentration.

Since $R = 1$ at equilibrium,

$$\frac{k_{\text{attach,close}}(0)}{k_{\text{attach,far}}(0)} = \frac{k_{\text{cleave,close}} k_{\text{CW}}}{k_{\text{cleave,far}} k_{\text{CCW}}}, \tag{8}$$

allowing the NESS R to be well approximated in terms of attachment rates alone:

$$R \approx R_{\text{approx}} = \frac{k_{\text{attach,far}}(\langle N_{\text{FTC}} \rangle)}{k_{\text{attach,far}}(0)} \frac{k_{\text{attach,close}}(0)}{k_{\text{attach,close}}(\langle N_{\text{FTC}} \rangle)}. \tag{9}$$

Figure 6a shows that adding fuel increases attachment rates, both attachment near and far from the shuttling ring, but the speed-ups are not equal. Because $k_{\text{attach,far}}$ increases more steeply than $k_{\text{attach,close}}$, $R > 1$ and current is CW. Our analysis of R shows that FC1, FC2, and FC3 all contribute to CW current, but FC1 contributes more strongly. By also monitoring the NESS population of the eight states (Fig. 6b), we show that increasing fuel takes population away from states 4 and 6, which lie on FC2 and FC3, but not FC1. The increase in CW bias with $\langle N_{\text{FTC}} \rangle$ in Fig. 3a can be viewed as a consequence of the fully ratcheted FC1 cycle becoming dominant.

We note that even with our analysis, it is not obvious how the geometry of the design in Fig. 1 translates into the CW currents. The fuel-dependent attachment rates both increase with FTC concentration, and directionality is determined by which of those rates rises up more rapidly with added $N_{\text{FTC}}$. We anticipate it will be possible to preserve the motor geometry and design changes to the motor's pairwise potential to yield $R < 1$ and CCW cycles.

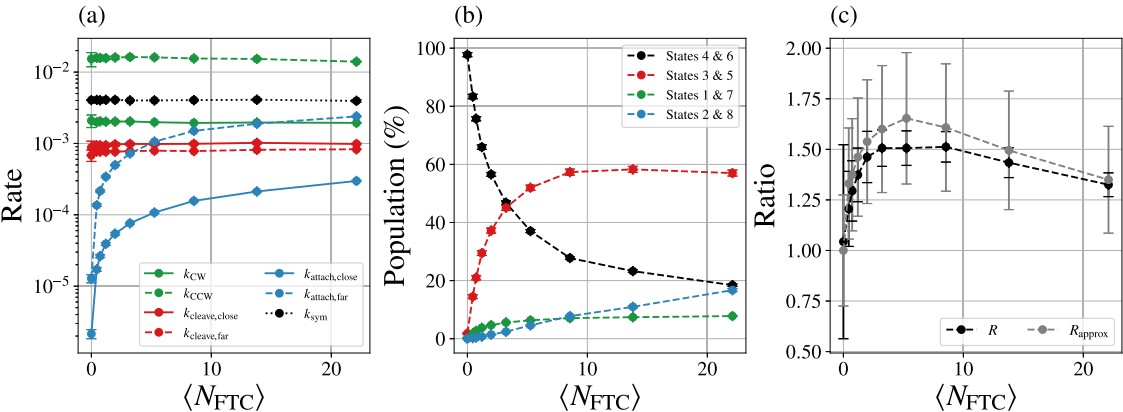

**Fig. 6 Parameterization of the kinetic model from simulation. a** The transition rates of the eight-state model from Fig. 5 and **b** populations for each of the eight states for Motor II as a function of the average number of FTC present in the simulation $\langle N_{FTC} \rangle$. **c** The ratio of rates $R(= e^{\mathcal{A}})$ and the approximate ratio from Eq. (9) as a function of FTC present. Data points show the average over 50 independent simulations of $t_{obs} = 1 \times 10^6$ units of time each. FTC number is varied by changing $\mu'_{FTC}$ from –10 and –1 to 1 in increments of 0.25. Data shown represent the mean and standard error from 50 independent simulations at each data point.

**Thermodynamics**. We have described an entirely kinetic analysis, a discussion that suffices to explain the motor's directionality. If one wants to understand the thermodynamic cost of fueling the motor, however, care must be taken in connecting the kinetics with thermodynamics. At the fine-grained level of the NESS simulations, each step is microscopically reversible—both the chemostat GCMC moves and the Langevin time steps may be executed in reverse, backtracking and undoing the forward dynamics. At this microscopic level, the ratio of probabilities of forward and reverse steps measures the increasing entropy of the ideal particle reservoirs. We caution, however, that the link connecting forward and reversed rates to thermodynamics is more complicated upon coarse graining the configuration space (e.g., into the 8-state kinetic model). In that picture, it becomes important that the forward and reversed transitions between course-grained states often proceed via distinct pathways.

To make this point more explicit, we elaborate upon the transitions between states 1 and 4, characterized simply by $k_{cleave,close}$ and $k_{attach,close}$ in Fig. 5. In equilibrium simulations with only C and no tetrahedral clusters, the pathways for cleavage and attachment are identical, but simulations with FTC reveal differing pathways for cleavage and attachment (see Supplementary Movies 2 and 3). A minimal model to address the motor's thermodynamics must separate the pathways into the equilibrium-like process mediated by the C reservoir and an additional pathway mediated by the FTC and ETC reservoirs.

In light of these distinct mechanisms, we note that the previously described affinities $\mathcal{A}$ are cycle affinities of the Markov model and not thermodynamic affinities, which relate to the entropy produced by the motors. That physical entropy production can dramatically exceed the Markov model's entropy production when the rates of distinct pathways are clumped together as in Fig. 5. Consider, for example, Fig. 7, which illustrates a refinement to the kinetic model that resolves whether cleavage and attachment events were mediated by C alone ($k_{attach}^C$ and $k_{cleave}^C$) or by a tetrahedral cluster reaction ($k_{attach}^{TC}$ and $k_{cleave}^{TC}$). The refinement does not alter the rate of shuttling ring current provided $k_{attach} = k_{attach}^C + k_{attach}^{TC}$ and $k_{cleave} = k_{cleave}^C + k_{cleave}^{TC}$. Though the current is insensitive to the refined model, the two Markov models produce entropy at different rates. Figure 7 Markov model includes additional cycles from state 1 to 4 and back via the other pathway, and the entropy production associated with those cycles is undetected by Fig. 5 model. In other words, coarse graining yields a model that produces less

entropy than the fine-grained model, a well-known effect of the data processing inequality that applies whether the coarse graining combines together microstates or pathways[54,55]. It is therefore notable that our simulations give access to the reversibility of the trajectories in the full state space, not just the reversibility of some reduced Markov models. We anticipate that capability will be particularly beneficial for future studies of the thermodynamic performance.

## Discussion

The models and methods presented here demonstrate a computational strategy to study how pairwise interactions give rise to dynamical function by simulating Langevin dynamics of a motor model simultaneously with GCMC chemostats. One can imagine carrying out similar, albeit vastly more expensive, simulations using more detailed, realistic models of chemical motors, but we highlight that our minimal toy model offers a tractable playground for exploring principles. It provides practical access to calculations of efficiency, accuracy, speed, and entropy production in a nontrivial particle-based model, opening the door to further explorations of thermodynamic and kinetic bounds[56,57] that limit what sort of autonomous, steady-state motors can be designed. Those studies of the interplay between fluctuations and dissipation are commonly applied to abstract nonequilibrium Markov jump models without explicitly specifying the microscopic origin of the rates. We anticipate that the stochastic thermodynamics community will benefit from this toy model that enables an explicit connection between pair potentials and the mesoscopic transition rates. We also anticipate that our approach will be useful in testing proposed improvements to the motor's design[58].

More concretely, our work should aid in the design and implementation of autonomous mesoscale machines. While this work was inspired by a molecular-scale motor[15], the pairwise potentials we use could more easily be built from mesoscale colloid constructions, where interactions between subunits can be tuned[59]. Significantly, we demonstrated that the motor maintains directional current in the overdamped regime, which is relevant to such colloidal diffusion. Although we do not expect our particular tetrahedral cluster fuel to be the most reasonable design on which to build an experimentally accessible mesoscale machine, we do hope the illustration and the methods will encourage more designs that will soon be experimentally realized.

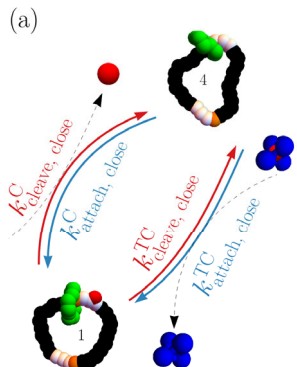

(a)

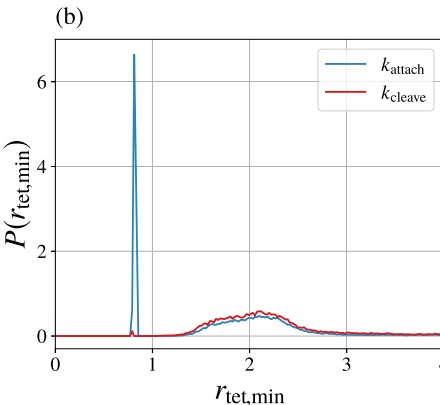

(b)

**Fig. 7 Examining distinct pathways in the kinetic model.** Thermodynamic interpretation requires separation into pathways, distinguished by their interplay with the reservoirs. **a** The attachment rate $k_{attach,close}$ to pass from state 1 to 4 in Fig. 5 is a superposition of two rates: $k^{C}_{attach,close}$, the rate for a free C to bind, and $k^{TC}_{attach,close}$, the rate for a C to be extracted from an FTC to create a blocking group. Even though each pathway is microscopically reversible, typical transformations from 1 to 4 are not time-reversals of the typical transformations from 4 to 1. The former prefers the C pathway while the latter prefers TC. **b** The distinction between forward and reversed pathways is detected by recording the distance between the C blocking group and the nearest tetrahedral structure at the time of the attachment and cleavage transitions. Combining data from both close and far events with $\mu'_{FTC} = 1$, $\mu'_{C} = -3$, and $\mu'_{ETC} = -10$, the distribution of distances $P(r_{tet,min})$ shows attachment events have a high probability of being mediated by a tetrahedral cluster evidenced by the large peak around the reaction distance of 0.8. By contrast, cleavage events have essentially no probability of happening via tetrahedral mediation.

## Methods

**Model details**. We used a modified Lennard–Jones (LJ) potential for all non-bonded interactions between particles in the system. Unlike the standard WCA potential[60], which includes both $r^{-12}$ and $r^{-6}$ contributions in the repulsive regime, we modified pairwise LJ potentials by introducing separate control over the strength of the $r^{-12}$ repulsive and $r^{-6}$ attractive terms:

$$U_{LJ}(\mathbf{r}_{ij}) = 4\epsilon_{R,ij}\left(\frac{\sigma_{ij}}{|\mathbf{r}_{ij}|}\right)^{12} - 4\epsilon_{A,ij}\left(\frac{\sigma_{ij}}{|\mathbf{r}_{ij}|}\right)^{6}, \tag{10}$$

where $\sigma_{ij}$ is the average of the radii of particles $i$ and $j$. The strength of the short-ranged repulsive interaction between particles $i$ and $j$ is tuned by $\epsilon_{R,ij}$, while that of the long-range attractive interaction is tuned by $\epsilon_{A,ij}$, as in[61]. All particles in the system are volume-excluding ($\epsilon_{R,ij} > 0$), but only some pairwise interactions are attractive ($\epsilon_{A,ij} \geq 0$). The full set of interaction parameters for each type of particle in the system is given in Supplementary Table 2.

The FTC fuel molecules are comprised of a four-particle tetrahedron bound along the edges (blue) and a free central particle (red), depicted in Fig. 1. The edges of the tetrahedron are held together with harmonic interactions that seek to minimize the distance $\mathbf{r}_{ij}$ between particles $i$ and $j$:

$$U_{harmonic}(\mathbf{r}_{ij}) = \frac{1}{2}k_{ij}\mathbf{r}_{ij}^{2}. \tag{11}$$

The values of the spring constants $k_{ij}$ are found in Supplementary Table 2. The particle types of the tetrahedron are labeled as TET1, TET2, TET3, and TET4, while the central particle is called CENT. Pairwise interactions between all of these particle types are purely repulsive ($\epsilon_{A,ij} = 0$). This ensures that FTC is a metastable, kinetically trapped state and it also ensures that FTC, ETC, and C do not aggregate in the simulation cell. Progress along the FTC → ETC + C reaction pathway is tracked by measuring $r$, the distance between the C particle and the center of mass of the four tetrahedron particles. In non-dimensional units, the cluster is in the FTC state when $r \leq 0.25$, it is in the ETC + C state when $r \geq 0.8$, and it is in an intermediate transition regime, visited fleetingly, when $0.25 < r < 0.8$.

The motor model is composed of two interlocked rings. A large ring consisting of $N_{large} = 30$ connected beads functions as a track for a smaller shuttling ring (green) with $N_{shuttle} = 12$ beads to diffuse or shuttle around, as depicted in Fig. 1. The shuttling ring is made up of a single-particle type, labeled SHUTTLE. The large ring is made up of three particle types: INERT particles that are purely volume-excluding (black), BIND particles that have attractive interactions with the shuttling ring (orange), and catalytic particles, labeled CAT1, CAT2, CAT3 (white), that have attractive interactions with TET1, TET2, TET3, TET4, and CENT particles to facilitate the decomposition of FTC to ETC + C. The ring is arranged so that a three-particle catalytic site (CAT2-CAT1-CAT3 in CW order) is on the CW side of a single-particle binding site, followed by a set of 11 inert particles before the binding/catalytic motif repeats on the opposite side of the large ring. The binding sites, located at large ring indices 0 and 15, are analogous to the fumaramide residues of the Wilson et al. motor[15]. The catalytic sites, located at large ring indices 1–3 and 16–18, are analogous to the hydroxy groups of the Wilson et al. motor. The attractive interaction between C (CENT) particles and the middle catalytic particle (CAT1) is particularly strong so as to hold the C particle near the catalytic site as a blocking group for the shuttling ring after a catalyzed

reaction has occurred. Those blocking groups are especially effective at preventing the diffusion of the shuttling ring because C particles also have particularly strong repulsions with the shuttling ring particles (SHUTTLE).

The rings have intramolecular interactions similar to those used for coarse-grained polymer models where bond and angle potentials maintain geometry and the modified LJ potential of Eq. (10) serves to include volume exclusion. The bonded interactions between adjacent beads in the motor rings is given by a finitely extensible nonlinear spring (FENE) potential:

$$U_{FENE}(\mathbf{r}_{ij}) = -\frac{1}{2}k_{F,ij}r_{max,ij}^{2}\log\left[1 - \left(\frac{|\mathbf{r}_{ij}|}{r_{max,ij}}\right)^{2}\right]. \tag{12}$$

Here $\mathbf{r}_{ij}$ is the displacement vector between particles $i$ and $j$, $k_{F,ij}$ is the FENE force constant, and $r_{max,ij}$ is the maximum extension between the particle pair. Groups of three adjacent ring particles also have angular interactions to maintain the overall circular geometry of the ring:

$$U_{angle}(\theta_{ijk}) = \frac{1}{2}k_{A,ijk}\left(\theta_{ijk} - \theta_{0,ijk}\right)^{2}, \tag{13}$$

where $i$ is the index of the middle particle of the three adjacent $i$, $j$, and $k$ particles, $k_{A,ijk}$ is the angular force constant, $\theta_{ijk}$ is the angle formed by the three particles, and $\theta_{0,ijk}$ is the equilibrium angle. For the shuttling ring $\theta_{0,ijk} = \pi\left(1 - \frac{2}{N_{shuttle}}\right)$ and for the large ring $\theta_{0,ijk} = \pi\left(1 - \frac{2}{N_{large}}\right)$. The bond and angle parameters as well as the modified LJ parameters for all of the motor particles are found in Supplementary Tables 2 and 3. The shuttling ring and large ring are placed in an interlocked configuration. No bonded (FENE or angular) interactions connect the two rings as they can be held in an interlocked state through the volume exclusion of the LJ interaction alone. The shuttling ring is therefore free to diffuse around the large ring.

**Method details**. To propagate the system dynamics forward in time we solve Eq. (2) numerically with a time step of $\Delta t = 5 \times 10^{-3}$ for some number of time steps $N_{steps}$ using the integrator of Athènes and Adjanor[62]:

$$\begin{aligned}
\mathbf{p}_i^{j+\frac{1}{2}} &= \mathbf{p}_i^j e^{-\frac{\gamma\Delta t}{2m_i}} + \mathbf{f}_i^j\frac{\Delta t}{2} + \boldsymbol{\eta}_i^{j+\frac{1}{2}} \\
\mathbf{r}_i^{j+1} &= \mathbf{r}_i^j + \mathbf{p}_i^{j+\frac{1}{2}}\frac{\Delta t}{m_i} \\
\mathbf{p}_i^{j+1} &= \left[\mathbf{p}_i^{j+\frac{1}{2}} + \mathbf{f}_i^{j+1}\frac{\Delta t}{2}\right]e^{-\frac{\gamma\Delta t}{2m_i}} + \boldsymbol{\eta}_i^{j+1},
\end{aligned} \tag{14}$$

where $\mathbf{f}_i = -\nabla U(\mathbf{r}_i)$ is the force on particle $i$, $\mathbf{r}_i^j \equiv \mathbf{r}_i(j\Delta t)$ is the position of particle $i$ at time $j\Delta t$, $\mathbf{p}_i^j \equiv \mathbf{p}_i(j\Delta t)$ is the momentum of particle $i$ at time $j\Delta t$, and each $\boldsymbol{\eta}_i$ is a random vector with components drawn from a zero-mean Gaussian with variance $m_i(1 - \exp(-\gamma\Delta t/m_i))k_B T$. Other choices of numerical integrator are possible[63]. We performed all simulations in non-dimensional form with characteristic length given by the LJ radius of an INERT particle, characteristic energy given by the repulsive strength of the INERT–INERT interaction, and characteristic mass given by the mass of an INERT particle. All of these values were then set to unity, i.e. ($\sigma_{INERT} = 1$, $m_{INERT} = 1$, $\epsilon_{R,INERT-INERT} = 1$), respectively. All other particle masses were also set to unity, and the only particles with non-unit radii were CENT

particles with $\sigma_{\text{CENT}} = 0.45$. We report data for simulations with $k_{\text{B}}T = 0.5$ and with $\gamma = 0.5$, except where otherwise noted. For completeness, the full set of particle mass and size parameters are given in Supplementary Table 1.

We performed a GCMC move every 100 Langevin time steps in order to maintain the system at a steady state concentration of FTC, ETC, and C. The GCMC moves were conditionally accepted so the chemostatted region of space would target the grand canonical distribution

$$P(\mathbf{r}, \mathbf{p}) = \frac{1}{\Xi} \frac{e^{\beta\mu_{\text{FTC}}N_{\text{FTC}} + \beta\mu_{\text{ETC}}N_{\text{ETC}} + \beta\mu_{\text{C}}N_{\text{C}} - \beta H(\mathbf{r}, \mathbf{p})}}{N_{\text{FTC}}! N_{\text{ETC}}! N_{\text{C}}!}, \quad (15)$$

where $\mathbf{r}$ and $\mathbf{p}$ are vectors of fluctuating length containing the coordinates for each copy of each species and $\Xi$ is the grand canonical partition function. The number of copies of each species ($N_{\text{FTC}}$, $N_{\text{ETC}}$, and $N_{\text{C}}$) can be viewed as functions of $\mathbf{r}$ and $\mathbf{p}$, as can the total energy $H(\mathbf{r}, \mathbf{p})$, the kinetic energy $K(\mathbf{p})$, and the potential energy $U(\mathbf{r})$. Though we are ultimately interested in unlabeled particles, it is simplest to utilize unphysical labels for accounting purposes. Marginalizing over all equally probable permutations of labels gives the density for unlabeled particles, which lacks the denominator of Eq. (14).

In practice, the GCMC method described here differs slightly from a standard implementation[18] since two of the species coupled to external chemical potentials (FTC and ETC) have internal degrees of freedom. Each GCMC chemostat move begins by randomly and uniformly selecting which of the three species to act on and whether to add or remove that species. The chemostat only acts on the outer volume of Fig. 1, and all copies of the chosen species occupying that outer volume are equally likely to be removed in the generated trial move. In the usual Metropolis manner, that trial removal of the copy of species $i$ is conditionally accepted with probability

$$P^{\text{acc}}_{i,\text{removal}}(\mathbf{r}, \mathbf{p} \to \mathbf{r}', \mathbf{p}') = \min\left[1, N_i(\mathbf{r}) e^{-\beta(U(\mathbf{r}') - U(\mathbf{r}) + U_i^0)} e^{-\beta(\mu_i - A_i^0)}\right], \quad (16)$$

where $U^0$ is the internal potential energy of the removed species, $Z_i^0$ is the canonical partition function for a single $i$ cluster in a box of volume $V_0$, and $A_i^0 = -k_{\text{B}}T \log Z_i^0$ is the associated free energy. In this work, we have operated in terms of the shifted chemical potential $\mu_i' \equiv \mu_i - A_i^0$ so the conditional acceptance probability was computed without needing to explicitly compute $A_i^0$ for the different cluster types. We tune these shifted chemical potentials from $\mu' = -10$ on the low end to $\mu' = 1$ on the high end.

The moves that add a cluster are more complicated because we must first generate a configuration of the cluster[52]. We used Monte Carlo to pre-generate an equilibrium ensemble of 10,000 configurations each of a single FTC cluster and of a single ETC cluster. An addition move first uniformly selects one of those Boltzmann-distributed configurations (a step which is moot when adding C). This configuration is randomly rotated in space then randomly inserted into the chemostatted volume. Velocities for the new particles are sampled from the Boltzmann distribution to complete the generation of trial coordinates $\mathbf{r}'$ and $\mathbf{p}'$. Analogous to the removal moves, the addition is conditionally accepted with probability

$$P^{\text{acc}}_{i,\text{addition}}(\mathbf{r}, \mathbf{p} \to \mathbf{r}', \mathbf{p}') = \min\left[1, \frac{1}{N_i(\mathbf{r}')} e^{-\beta(U(\mathbf{r}') - U(\mathbf{r}) - U_i^0)} e^{\beta(\mu_i - A_i^0)}\right]. \quad (17)$$

One confirmation that all three chemostats simultaneously function as desired is the demonstration of ideality in the dilute limit, discussed further in the SI.

These GCMC moves only occur in the space between the inner and outer simulation boxes, depicted in Fig. 1. Our simulation boxes were concentric cubes with inner side length $L_{\text{inner}} = 30$ and an outer cube of side length $L_{\text{outer}} = 34$. The motor itself is confined to the inner simulation box so that its dynamics are not directly perturbed by abrupt GCMC insertions and deletions. The motor is confined to the inner box with a LJ wall potential:

$$U_{\text{wall}}(\mathbf{r}_i) = 4\epsilon_{\text{wall}} \sum_{\alpha=x,y,z}\left[\left(\frac{\sigma_{\text{wall}}}{r_{\alpha,i} - \frac{1}{2}L_{\text{inner}}}\right)^{12} + \left(\frac{\sigma_{\text{wall}}}{r_{\alpha,i} + \frac{1}{2}L_{\text{inner}}}\right)^{12}\right], \quad (18)$$

where $\mathbf{r}_i = (r_{x,i}, r_{y,i}, r_{z,i})$ is the position of the $i$th motor particle and both boxes are centered at the origin. We set $\epsilon_{\text{wall}} = 1$ and $\sigma_{\text{wall}} = 1$. Particles of the FTC, ETC, and C molecules do not experience this wall potential and move freely between the inner and outer boxes. These species are also free to pass through the periodic boundaries of the outer box, which we implemented using the minimum image convention[18].

## Data availability

The data generated in this study have been deposited in a Zenodo.com repository under accession code https://doi.org/10.5281/zenodo.4481182. Data are available for Figs. 2, 3, 4, 6, and 7.

## Code availability

The code used in this study has been deposited in a Zenodo.com repository under accession code https://doi.org/10.5281/zenodo.4481182.

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

## Acknowledgements

The authors gratefully acknowledge productive conversations with Hadrien Vroylandt, Geyao Gu, and Rueih-Sheng Fu. Research reported in this publication was supported in part by the International Institute for Nanotechnology at Northwestern University and in part by the Gordon and Betty Moore Foundation through Grant No. GBMF10790.

## Author contributions

A.A. and T.R.G. jointly designed the study, conducted the simulations, analyzed the data, and prepared the manuscript.

## Competing interests

The authors declare no competing interests.
