## [Peer Review File · Nature Communications]

REVIEWER COMMENTS

Reviewer #1 (Remarks to the Author):

Referee report NCOMMS-21-07274

A. Albaugh and T. R. Gingrich, "Simulating a Chemically-Fueled Molecular Motor with Nonequilibrium Molecular Dynamics"

The authors study the dynamics and energetics of a molecular motor using molecular dynamics simulations. The model is simplistic in the sense that it tries to replicate the behaviour fuelled motor enzymes with a minimal number of particles that interact via simple potentials, yet it is significantly more detailed than the coarse-grained Markov models commonly used in stochastic thermodynamics. Non-equilibrium conditions are maintained via chemostats that regulate the concentration of fuel and product molecules outside of the space accessible to the motor. Finally, the model is coarse-grained to an eight-state Markov rate model, with rates determined from the simulation.

I think this manuscript presents excellent work, as it bridges the scales relevant for biomolecular chemistry (exemplified here with the molecular motor of Ref. [5]) with the coarse-grained Markov models commonly used in biophysics. It should therefore be of interest to both communities. Below, I list a number of questions and comments on aspects that appeared somewhat ill-presented to me (in roughly chronological order). If these can be adequately answered, I am sure that I will be able to recommend an accordingly revised manuscript for publication in Nature Communications.

1.

1st paragraph: Thermal fluctuations are first presented as a "basic ingredient" which microscopic machines rectify, a sentence later they appear more like a nuisance that lead to challenges in the design. I think both pictures are part of the truth, but they appear somewhat contradictory here. Maybe a better formulation can be found (or this ambivalence be commented on).

2.

I. 96: spell out ADP, P

3.

After reading the section "A Classical Motor Model" and the caption of Fig. 1, I didn't quite understand how the motor actually works, e.g., I couldn't figure out in which direction it would move. These questions I only found answered with the rate model later (which maybe could be referred to here). Would it be fair to say that, at this point, the motor does not follow any specific design, other than the asymmetry and the coupling to the non-equilibrium reaction as necessary requirements for directed motion? Or is there a more concrete motivation for this design (other than the motor of Ref. [5] it is modeled after)?

4.

In Eq. (4), add $\dot{x}=p/m$ for completeness.

5.

Is there a specific reason for the choice of the integrator of Eq. (5)? Would it sample the Boltzmann distribution exactly in the absence of chemical driving?

6.

Why do the GCMC trial moves occur periodically every 100 time steps? I would have found it more natural to have them at stochastic points in time. I don't think it's an issue here (100 time steps is surely short enough to be considered continuous), but in principle this could lead to a periodically driven state rather than a NESS.

7.

l. 252 "... Metropolis factor that depends on the chemical potentials of the three species." I find this somewhat confusing. I assume the chemical potentials are the chemostatted ones (not some instantaneous quantities measured somehow in non-equilibrium), and therefore constant. But the Metropolis factor is surely not constant, so it should also be mentioned that it depends on the configuration/concentration.

8.

l. 257: strange formulation, not sure whether the convenience can be regarded as "consequence"

9.

l. 286: "relative to equilibrium": Are you comparing to a state completely without FTC/ETC/C or with equilibrium concentrations thereof (or does it not matter)?

10.

The definition of accuracy might be misleading. A motor that always takes two forward steps followed by one backward step at precise points in time would still be very accurate in the long run. The effective diffusion coefficient of the molecular motor (as considered in the thermodynamic uncertainty relation) would surely be a better quantification of accuracy, which may be harder to measure in short simulation runs. Anyway, the definition of accuracy and its motivation is made clear in the text.

11.

The definition of efficiency is somewhat at hoc, it should be acknowledged that more sophisticated measures for the efficiency of molecular motors have been established [e.g. that of Derenyi et al, PRL 83, 903 (1999)]. I find it problematic that the input is just quantified as the total number of catalysed reactions, regardless of the chemical potential difference. A machine that produces the same output with the same number of catalysed reactions but higher chemical potential difference is surely less efficient. At least the net number of chemical reactions should be considered for the input, i.e. with backward reactions subtracted, which would be zero in equilibrium (though it seems this can be neglected in practice because backward reactions are unlikely). Thermodynamically, efficiency would be considered in terms of the overall input of free energy and the work produced against a loading force, which might be hard to realise in the simulation though. But some kind of Stokes efficiency could be considered using the diffusivity of the shuttle in the absence of C. The simple definition chosen by the authors may serve well for this paper, but in future this molecular toy model could serve well to compare various definitions of efficiency.

12.

Is there an intuitive explanation why the accuracy increases in the overdamped regime? And what would happen to the current if the overdamped regime was reached by reducing the particle mass rather than by increasing gamma? That way it might be easier to pinpoint the role of inertia.

13.

What was the motivation to show in Fig. 3 the mean fuel concentration rather than the (controlled) chemical potential on the x-axis? A reminder of what $\langle N_{FTC} \rangle$ is would help in the figure caption.

14.

I.439: Check grammar: "faster [...] than a proximal site" doesn't make sense

15.

I think a better discussion of the symmetry of the rate model is needed, I see some issues with the current presentation. I could not find an explanation of the rates k_{sym} anywhere in the text. Do they describe clockwise and/or counterclockwise motion of the shuttle along the ring (I assume they do not describe a half-rotation of the whole ring with the shuttle fixed)? If one distinguishes the sense of rotation (as necessary to evaluate e.g. the accuracy), the model gets one more fundamental cycle, corresponding to the spontaneous, non-driven rotation of the motor while no C is bound. I personally would have favoured a simplified, yet fully equivalent four-state representation of the model, which identifies states that are equivalent up to a half-rotation. Rather than characterising the state of the binding sites by their absolute position, one could simply label the binding sites relative to the shuttle position (one in front and one behind the shuttle). The " k_{sym} " transitions (in CW or CCW direction) would then lead from the state with no bound C into itself, which might be confusing for the reader (but in my opinion not more confusing than the issue with these transitions in the current presentation).

16.

The molecular toy model offers probably more potential for comparison with the rate model than what is in the scope of the present manuscript. I assume that, by construction, the rate model matches the steady-state distribution and the average motor current of the molecular model. But one could also ask other questions, e.g. how well the rate model would predict fluctuations of the current or the distribution of time spent in coarse grained states.

17.

I. 452: Refer to Fig. 6 again as an explanation of what FC1 means.

Reviewer #2 (Remarks to the Author):

Review attached.

This paper demonstrates coarse-grained simulations of a molecular motor driven in an open system. The authors explicitly simulated chemical transformations of fuel molecules, and show that the interaction of the by-product of fuel molecule can ratchet the dynamics of a catenane-like molecular system into a preferred direction. The chemical reaction of the fuel is modeled such that a particle trapped inside a tetrahedral cluster is decomposed from the cluster when the particle-containing cluster is perturbed upon interaction with catalytic site of the shuttling ring. The product (C particle) released from the tetrahedral cluster serves as a group blocking the diffusion of the shuttling ring.

The model proposed in this manuscript is an explicit simulation of Markov jump processes on a network in nonequilibrium steady state. By tuning the rate constants of the network by means of chemical potentials of fuel molecules and/or strengths of inter-particle interactions, non-equilibrium currents, physically corresponding to uni-directional motion of molecular motor, can be generated. This is conceptually very clear and has been studied extensively in recent years. The authors' self-appraisal on the significance of the study is: "This work identifies inter-particle interactions that tune those rates to create a functional motor, thereby identifying a computational playground to investigate the interplay between accuracy, current generation, and efficiency of molecular information ratchet."

Let me summarize my overall assessment of this paper.

This manuscript is a proof-of-concept type of study, which presents a multi-scale modeling/simulation of molecular motor in nonequilibrium steady state with the chemical reaction of molecular fuel being explicit. However, (1) Statistical physical aspect of the nonequilibrium Markov dynamics on a network is already well studied. (2) Controlling the stabilities and/or kinetic rates of reaction processes via synthesis of molecules or functionalization on the already existing molecules pertains to the realm of long-standing research activities in chemistry, protein sciences, and engineering. Therefore, the idea of tuning the kinetic rates by means of inter-particle interactions is not entirely new. (3) The chemical reaction of the molecular fuel model (from JCP 153, 204102 (2020)) used in this paper is interesting, but it does not appear to be well aligned with those molecular fuels consumed in real processes (see my detailed comment #4).

In multiple places, the paper is rather loosely written. For example, some panels are described only briefly in the figure caption, but without in-depth explanation, which enforces readers to make guesses what each statement means by jumping back and forth across the manuscript. I also feel that the description of model, method, and results are mixed together, which renders the model description of Catenane-like molecular motor scattered all over the place. Especially, some of the contents (e.g., energy potentials for the fuel model in "A classical fuel model", integrator for the equation of motion and wall potential in "Dynamics") which better fit to Method section, are demonstrated in the Results section. Due to this presentation style, it is not easy to digest the current form of the paper. The authors should make some effort to clean up the manuscript so that it reads more smoothly.

The citations in this paper are extremely biased. Out of 72 citations, 38 papers (more than 50 %) are from two groups: Kapral (24) and Astumian (14). Also, the authors' scope on real biological motors are narrow, and only limited to ATP synthase. In light of the diversity of the molecular motor field, I find it unacceptable and hard to justify.

More detailed comments:

1. The authors should explain the molecular mechanism of CW (or CCW) rotation by words in one paragraph. I understand that C-particle, dissociated from the tetrahedral cluster, interacts with the ring polymer and blocks the diffusion of the shuttling ring, but this alone does not explain the biased (or ratcheted) motion. Even after reading the whole manuscript, it is unclear how unidirectional rotation is generated from the system. Fig.6 and accompanying explanation in Page 7 do not help. They are not comprehensive. I know that Ref. [5] by Leigh and colleagues is mentioned, but the paper should be self-contained.

2. The authors are advised to make the definitions of *accuracy* and *efficiency* mathematically more explicit and clearer.

(1) It is not clear what “The fraction of cycles in the CW direction” means. Is it $CW/(CW+CCW)$?

(2) “The net number of cycles (CW is positive, CCW is negative) per catalyzed reaction”

(3) There is also a comment on the “current” when discussing the trade-offs between accuracy, efficiency, and current. Please define the current.

3. On page 4, it says “The right panels of Fig.2 show that Motor II is able to generate more current, not because it consumes more fuel, but rather *because it more efficiently converts that fuel consumption into directed motion ...*”

(1) Which panels ... (b) and (e) or (b), (c), (e), (f)?

(2) It is not immediately clear where the italicized judgement originates from.

I can guess that the panels (b), (c), (e), (f) are being referred to, and that the judgement comes from the fact that the current of MODEL II is greater despite the fact that the fuel consumption is identical for MODEL I and II based on Fig.2 (c) and (f). But, these are not explained explicitly.

4. Fig.2 (c) and (f) show the number of reaction (decomposition of the reactive particle from the tetrahedral cluster) for MODEL I and II. These panels appear to indicate that significant fraction (3/4) of fuel molecules decompose spontaneously without “catalysis”, and only 1/4 of them are catalyzed and decomposed into products. Apparently, 3/4 of the fuel molecules undergo futile consumption. I’ve never heard such a molecular fuel in real biological systems.

5. “*In a tightly coupled regime* one might expect an efficiency of 0.5 (one net cycle for every two catalyzed reactions).” — What does it mean by a tightly coupled regime? and where this expectation of efficiency of 0.5 comes from?

6. The caption of Fig.3 says that the chemical potential μ'_{FTC} was varied, but then the x-axis is plotted against $\langle N_{FTC} \rangle$. I understand that Eq.(A9) relates these two quantities. But, if this is the author’s preferred way of presenting their result, the relation between the two quantities should be clarified or Eq. (A9) should be referred to in the text.

7. The subsection “Timescales” aims to show that the chemostat relaxation time is much faster than that of motor dynamics. First, the authors are advised to amend the title to be more specific. Second, please explain why there are the small dips when chemical potential is switched from that of equilibrium to nonequilibrium.

8. The authors discuss in “the eight state rate model,” indicating that while the R (affinity) is identical for the Motor I and II, the two models differ in terms of their CW/CCW current, efficiency, accuracy ..., thus pointing that one model is better than the other. Despite the same R, the blocked population is greater for Motor II (Fig.7). Since R is the ratio of all forward rates to all backward rates, there are multiple ways of getting the same R value, which is not at all mysterious.

I believe that the authors can demystify the difference of the two Motors by quantifying all the first-order rate constants for transition or all the reaction currents flowing along the edges of network for the Motor I and Motor II completely.

Dear Referees,

Thank you for your consideration of our manuscript “Simulating a Chemically-Fueled Molecular Motor with Nonequilibrium Molecular Dynamics”. Both reviews were particularly constructive and helped us craft an improved paper that we wish to resubmit. We have taken the past few months to make substantial changes to the manuscript in response to your feedback. We have clarified the definitions of accuracy, current, and efficiency. Even more significantly, we now have a thorough explanation of the underlying working principle that explains why the motor rotates clockwise. We thank the reviewers for nudging us to do additional work to clarify the mechanism. One of you found our structure challenging, so we have rearranged the model and method descriptions. We feel the new version is more intuitive and better fits the Nature Communications format.

We attach point-by-point responses below and a redline version to highlight the many rearrangements and improvements. We hope you enjoy this revised version and find it suitable for Nature Communications.

Regards,

Alex Albaugh and Todd R. Gingrich

Review 1: The authors study the dynamics and energetics of a molecular motor using molecular dynamics simulations. The model is simplistic in the sense that it tries to replicate the behaviour of fuelled motor enzymes with a minimal number of particles that interact via simple potentials, yet it is significantly more detailed than the coarse-grained Markov models commonly used in stochastic thermodynamics. Non-equilibrium conditions are maintained via chemostats that regulate the concentration of fuel and product molecules outside of the space accessible to the motor. Finally, the model is coarse-grained to an eight-state Markov rate model, with rates determined from the simulation.

I think this manuscript presents excellent work, as it bridges the scales relevant for biomolecular chemistry (exemplified here with the molecular motor of Ref. [5]) with the coarse-grained Markov models commonly used in biophysics. It should therefore be of interest to both communities. Below, I list a number of questions and comments on aspects that appeared somewhat ill-presented to me (in roughly chronological order). If these can be adequately answered, I am sure that I will be able to recommend an accordingly revised manuscript for publication in Nature Communications.

We thank the reviewer for their kind summary of our work and hope that the changes described will be satisfactory.

1. 1st paragraph: Thermal fluctuations are first presented as as “basic ingredient” which microscopic machines rectify, a sentence later they appear more like a nuisance that lead to challenges in the design. I think both pictures are part of the truth, but they appear somewhat contradictory here. Maybe a better formulation can be found (or this ambivalence be commented on).

We agree that both pictures are part of the truth but that the apparent contradiction is distracting in an introductory paragraph. We have removed the language about fluctuations being a challenge to the design.

2. l. 96: spell out ADP, P

The text now reflects this change.

3. After reading the section “A Classical Motor Model” and the caption of Fig. 1, I didn’t quite understand how the motor actually works, e.g., I couldn’t figure out in which direction it would move. These questions I only

found answered with the rate model later (which maybe could be referred to here). Would it be fair to say that, at this point, the motor does not follow any specific design, other than the asymmetry and the coupling to the non-equilibrium reaction as necessary requirements for directed motion? Or is there a more concrete motivation for this design (other than the motor of Ref. [5] it is modeled after)?

Yes, it's fair to say that when we first introduce the motor it is not obvious which direction it will turn. It is clear that an asymmetry is built into the design such that detailed balance can be broken, but the direction of those fluxes is quite subtle. We have added to lines 205-06 "a point we return to in a more detailed discussion of the mechanism." We have furthermore developed a significantly clearer demonstration of how the directionality relates to the cycle affinity and rates, which we present in the new section titled "Clockwise Directionality" starting on line 399.

4. In Eq. (4), add $\dot{\mathbf{x}} = \mathbf{p}/m$ for completeness.

The text now reflects this change.

5. Is there a specific reason for the choice of the integrator of Eq. (5)? Would it sample the Boltzmann distribution exactly in the absence of chemical driving?

As in ordinary MD, there are many choices of integrators, and the distinctions between them become less significant as the timestep decreases. For underdamped Langevin dynamics, a particularly thorough comparison of various integrators is described in Ref. 80, which we now point to in the main text, lines 661-62. We chose the particular integrator because, akin to symplectic leapfrog MD integrators, the symmetry between time-forward and time-reversed equations of motion is particularly appealing.

6. Why do the GCMC trial moves occur periodically every 100 time steps? I would have found it more natural to have them at stochastic points in time. I don't think it's an issue here (100 time steps is surely short enough to be considered continuous), but in principle this could lead to a periodically driven state rather than a NESS.

This is a very good point. Yes, in principle by scheduling GCMC steps every 100 time steps we are really setting up a time-periodically driven system. The referee's suggestion of stochastically choosing the time of the next GCMC move is a good one that we will incorporate into future work. However, as the referee seems to acknowledge, the GCMC moves are so frequent that the GCMC moves are actually quite decoupled from the motor. In other words, we determined that performing a GCMC move every 100 time steps is sufficiently frequent that the outer box is held in the correct steady state. In fact, this steady state could probably be maintained with a lower GCMC frequency, but we chose every 100 time steps to be conservative. The GCMC moves occur in an outer box to which the motor has no access and the species involved with GCMC must then diffuse to the motor. This diffusion could be thought of as effectively implementing some stochasticity to the time between events for when molecules hit the inner box, akin to what the reviewer suggests.

7. l. 252 "... Metropolis factor that depends on the chemical potentials of the three species." I find this somewhat confusing. I assume the chemical potentials are the chemostatted ones (not some instantaneous quantities measured somehow in non-equilibrium), and therefore constant. But the Metropolis factor is surely not constant, so it should also be mentioned that it depends on the configuration/concentration.

The reviewer is correct. The Metropolis factor is not constant (we did not claim it was), but does depend on the fixed external chemical potential and instantaneous concentrations within the simulations. We have updated the text to make this more clear: "These moves are conditionally accepted with a Metropolis factor that depends on the set chemical potentials of the three species and their instantaneous concentrations."

8. l. 257: strange formulation, not sure whether the convenience can be regarded as "consequence"

We agree. The sentence now reads: “Due to those internal degrees of freedom, the GCMC acceptance probabilities directly depend on $\mu' \equiv \mu - A^0$, the applied external chemical potential less the standard state Helmholtz free energy.”

9. 1. 286: “relative to equilibrium”: Are you comparing to a state completely without FTC/ETC/C or with equilibrium concentrations thereof (or does it not matter)?

We are referring to an equilibrium with no FTC, ETC, or C present. We have clarified the text to be more specific: “The NESS fuel concentration only slightly alters the distribution of the motor configurations relative to thermal equilibrium with no FTC, ETC, or C present.” In the new draft, however, we find it useful to also consider comparisons with an equilibrium that has C present, starting at line 399 in the “Clockwise Directionality” section.

10. The definition of accuracy might be misleading. A motor that always takes two forward steps followed by one backward step at precise points in time would still be very accurate in the long run. The effective diffusion coefficient of the molecular motor (as considered in the thermodynamic uncertainty relation) would surely be a better quantification of accuracy, which may be harder to measure in short simulation runs. Anyway, the definition of accuracy and its motivation is made clear in the text.

Indeed it is tricky to discuss both accuracy and efficiency (current is more straightforward) because both require a decision about what the goal of the process is. We have reworked the section beginning at line 253, particularly around Eqs. (3) - (5) to improve clarity.

11. The definition of efficiency is somewhat ad hoc, it should be acknowledged that more sophisticated measures for the efficiency of molecular motors have been established [e.g. that of Derenyi et al, PRL 83, 903 (1999)]. I find it problematic that the input is just quantified as the total number of catalysed reactions, regardless of the chemical potential difference. A machine that produces the same output with the same number of catalysed reactions but higher chemical potential difference is surely less efficient. At least the net number of chemical reactions should be considered for the input, i.e. with backward reactions subtracted, which would be zero in equilibrium (though it seems this can be neglected in practice because backward reactions are unlikely). Thermodynamically, efficiency would be considered in terms of the overall input of free energy and the work produced against a loading force, which might be hard to realise in the simulation though. But some kind of Stokes efficiency could be considered using the diffusivity of the shuttle in the absence of C. The simple definition chosen by the authors may serve well for this paper, but in future this molecular toy model could serve well to compare various definitions of efficiency.

Thank you for these productive comments. In the revised draft we clarify that we are measuring the “coupling efficiency, meaning the fraction of chemical reactions that yield a net clockwise cycle.” This object is different from the efficiency measures the reviewer mentions, which should be the focus of future work.

12. Is there an intuitive explanation why the accuracy increases in the overdamped regime? And what would happen to the current if the overdamped regime was reached by reducing the particle mass rather than by increasing γ ? That way it might be easier to pinpoint the role of inertia.

We don't presently have an intuitive understanding, but intend to more thoroughly consider the role of damping in future work.

13. What was the motivation to show in Fig. 3 the mean fuel concentration rather than the (controlled) chemical potential on the x-axis? A reminder of what $\langle N_{\text{FTC}} \rangle$ is would help in the figure caption.

We chose to use fuel concentration instead of chemical potential for ease of interpretation. Referencing performance relative to the number of fuel molecules instead of a chemical potential, in our opinion, gives the reader a more intuitive sense of what is happening in the simulations. Of course either perspective is valid and they are related through a concrete relationship, which we now point out in the Fig. 3 caption. We have also defined $\langle N_{\text{FTC}} \rangle$, as the reviewer suggested.

14. l.439: Check grammar: “faster [...] than a proximal site” doesn’t make sense

That section was removed in the rewritten manuscript.

15. I think a better discussion of the symmetry of the rate model is needed, I see some issues with the current presentation. I could not find an explanation of the rates k_{sym} anywhere in the text. Do they describe clockwise and/or counterclockwise motion of the shuttle along the ring (I assume they do not describe a half-rotation of the whole ring with the shuttle fixed)? If one distinguishes the sense of rotation (as necessary to evaluate e.g. the accuracy), the model gets one more fundamental cycle, corresponding to the spontaneous, non-driven rotation of the motor while no C is bound. I personally would have favoured a simplified, yet fully equivalent four-state representation of the model, which identifies states that are equivalent up to a half-rotation. Rather than characterising the state of the binding sites by their absolute position, one could simply label the binding sites relative to the shuttle position (one in front and one behind the shuttle). The “ k_{sym} ” transitions (in CW or CCW direction) would then lead from the state with no bound C into itself, which might be confusing for the reader (but in my opinion not more confusing than the issue with these transitions in the current presentation).

We have added a more detailed discussion of the eight-state model, and k_{sym} is now defined in the text on line 354. It is true that 4 of the states of the model are symmetrically equivalent to 4 other states and this symmetry is used when we collect statistics to parameterize the model’s rate constants. We have chose to leave the model depicted as is, though, because it gives the reader a picture of what an entire cycle looks like. The reviewer is correct that either representation presents drawbacks, but we prefer the depiction as is.

16. The molecular toy model offers probably more potential for comparison with the rate model than what is in the scope of the present manuscript. I assume that, by construction, the rate model matches the steady-state distribution and the average motor current of the molecular model. But one could also ask other questions, e.g. how well the rate model would predict fluctuations of the current or the distribution of time spent in coarse grained states.

The eight-state rate model can accurately capture the steady state populations and directionality of the simulations. We use it as a tool for understanding the motor’s mechanism, which is done through analysis of the rates. While we could examine questions of fluctuations as the reviewer suggests, we do not think it would contribute to the conclusions of this study. Overall we are trying to describe how simulation methodology can be used to create a NESS and how a motor model can successfully harness the free energy gradient of such a state to create directed motion. Questions of then mapping those dynamics to more approximate models do not contribute to that goal beyond helping to understand mechanisms, which we have done. We will note that the reviewer has once again hit upon a core idea behind on-going work. We are currently working to systematically build more principled coarse-grained models and understand how the level of coarse-graining affects the resultant thermodynamics and kinetics.

17. l. 452: Refer to Fig. 6 again as an explanation of what FC1 means.

We have incorporated this change into the text along with a more thorough discussion of fundamental cycles.

Review 2: This paper demonstrates coarse-grained simulations of a molecular motor driven in an open system. The authors explicitly simulated chemical transformations of fuel molecules, and show that the interaction of

the by-product of fuel molecule can ratchet the dynamics of a catenane-like molecular system into a preferred direction. The chemical reaction of the fuel is modeled such that a particle trapped inside a tetrahedral cluster is decomposed from the cluster when the particle-containing cluster is perturbed upon interaction with catalytic site of the shuttling ring. The product (C particle) released from the tetrahedral cluster serves as a group blocking the diffusion of the shuttling ring.

The model proposed in this manuscript is an explicit simulation of Markov jump processes on a network in nonequilibrium steady state. By tuning the rate constants of the network by means of chemical potentials of fuel molecules and/or strengths of inter-particle interactions, non-equilibrium currents, physically corresponding to uni-directional motion of molecular motor, can be generated. This is conceptually very clear and has been studied extensively in recent years. The authors' self-appraisal on the significance of the study is: "This work identifies inter-particle interactions that tune those rates to create a functional motor, thereby identifying a computational playground to investigate the interplay between accuracy, current generation, and efficiency of molecular information ratchet."

Let me summarize my overall assessment of this paper.

This manuscript is a proof-of-concept type of study, which presents a multi-scale modeling/simulation of molecular motor in nonequilibrium steady state with the chemical reaction of molecular fuel being explicit. However, (1) Statistical physical aspect of the nonequilibrium Markov dynamics on a network is already well studied. (2) Controlling the stabilities and/or kinetic rates of reaction processes via synthesis of molecules or functionalization on the already existing molecules pertains to the realm of long-standing research activities in chemistry, protein sciences, and engineering. Therefore, the idea of tuning the kinetic rates by means of inter-particle interactions is not entirely new. (3) The chemical reaction of the molecular fuel model (from JCP 153, 204102 (2020)) used in this paper is interesting, but it does not appear to be well aligned with those molecular fuels consumed in real processes (see my detailed comment #4).

We thank the referee for the thorough reading and analysis of our work. We agree that it is the synthesis of (1), (2), and (3) that carries the most value.

In multiple places, the paper is rather loosely written. For example, some panels are described only briefly in the figure caption, but without in-depth explanation, which enforces readers to make guesses what each statement means by jumping back and forth across the manuscript. I also feel that the description of model, method, and results are mixed together, which renders the model description of Catenane-like molecular motor scattered all over the place. Especially, some of the contents (e.g., energy potentials for the fuel model in "A classical fuel model", integrator for the equation of motion and wall potential in "Dynamics") which better fit to Method section, are demonstrated in the Results section. Due to this presentation style, it is not easy to digest the current form of the paper. The authors should make some effort to clean up the manuscript so that it reads more smoothly.

Thank you for the feedback. We have made many significant changes (see the redline version) to tighten and improve the manuscript. These changes include improved figure captions, now expanded with more detail. In direct response to this comment, we have also implemented a major rearrangement of the sections to more consistently group topics. In particular, the body features a high-level description of our model and approach, while most of the methodological details can now be found in the Methods section.

The citations in this paper are extremely biased. Out of 72 citations, 38 papers (more than 50 %) are from two groups: Kapral (24) and Astumian (14). Also, the authors' scope on real biological motors are narrow, and only limited to ATP synthase. In light of the diversity of the molecular motor field, I find it unacceptable and hard to justify.

We appreciate the feedback, which reflects how deeply we have engaged with the Astumian and Kapral works. In our revision we have included 8 additional citations ([19], [59], [65], [66], [71], [72], [75], [80]), none of which come from Kapral or Astumian. Some of these include additional references to real biological motors, though we note that this paper really is not intended to be a review of such motors. We certainly

welcome specific suggestions of work that the reviewer believes we have inappropriately omitted.

More detailed comments:

1. The authors should explain the molecular mechanism of CW (or CCW) rotation by words in one paragraph. I understand that C-particle, dissociated from the tetrahedral cluster, interacts with the ring polymer and blocks the diffusion of the shuttling ring, but this alone does not explain the biased (or ratcheted) motion. Even after reading the whole manuscript, it is unclear how unidirectional rotation is generated from the system. Fig. 6 and accompanying explanation in Page 7 do not help. They are not comprehensive. I know that Ref. [5] by Leigh and colleagues is mentioned, but the paper should be self-contained.

The reviewer's comment is very helpful. If we went through the trouble of making a pared down minimal model, shouldn't we at least be able to clearly explain how/why it works?! The comment inspired us to better elucidate the mechanism behind the directed motion, and we believe the new section "Clockwise Directionality" makes the paper substantially better. In that section, we show how to use an equilibrium reference to explain the directionality in terms of just the two attachment rates (close and far). Specifically, we show that the clockwise motion is explained by the difference in slopes of those two rates with respect to added FTC concentration.

2. The authors are advised to make the definitions of accuracy and efficiency mathematically more explicit and clearer. (1) It is not clear what "The fraction of cycles in the CW direction" means. Is it $CW/(CW+CCW)$? (2) "The net number of cycles (CW is positive, CCW is negative) per catalyzed reaction" (3) There is also a comment on the "current" when discussing the trade-offs between accuracy, efficiency, and current. Please define the current.

Thank you for this suggestion. We believe the rewritten section entitled "Accuracy, Current, and Coupling Efficiency" is now much clearer.

3. On page 4, it says "The right panels of Fig. 2 show that Motor II is able to generate more current, not because it consumes more fuel, but rather because it more efficiently converts that fuel consumption into directed motion ... " (1) Which panels ... (b) and (e) or (b), (c), (e), (f)? (2) It is not immediately clear where the italicized judgment originates from. I can guess that the panels (b), (c), (e), (f) are being referred to, and that the judgment comes from the fact that the current of MODEL II is greater despite the fact that the fuel consumption is identical for MODEL I and II based on Fig.2 (c) and (f). But, these are not explained explicitly.

We are sorry this point was not sufficiently clear. One difficulty is that we presented so many subfigures that had tremendous redundancy. We decided that showing the Motor I and II behavior in Fig. 2 does more harm than good. Rather, the comparisons between the motors are more easily achieved by Figs. 3 and 4. We now use Fig. 2 to make a simpler, clearer point: "Figure 2 also reflects two important manners in which the present model motor differs from biological machines like ATP synthase. Firstly, our motor is fairly loosely coupled—Figs. 2b and 2c show that a single net cycle requires roughly 35 catalyzed $FTC \rightarrow ETC + C$ reactions. Secondly, the model fuel is not as metastable as ATP. Even in the absence of a motor's catalytic site, FTC can degrade on simulation timescales. As such, Fig. 2c distinguishes between catalyzed decompositions that occur in proximity to the catalytic sites and the total decompositions that could occur elsewhere."

4. Fig.2 (c) and (f) show the number of reaction (decomposition of the reactive particle from the tetrahedral cluster) for MODEL I and II. These panels appear to indicate that significant fraction (3/4) of fuel molecules decompose spontaneously without "catalysis", and only 1/4 of them are catalyzed and decomposed into products. Apparently, 3/4 of the fuel molecules undergo futile consumption. I've never heard such a molecular fuel in real biological systems.

The reviewer’s deduction is correct, and as mentioned in the previous point we are now highlighting this fact directly in the text. We, too, think that fuels in biological systems tend to have much longer lifetimes such that the catalyzed decomposition is far more prevalent than uncatalyzed. This system is “leaky” in the sense that it is losing out on some of the driving force through the uncatalyzed reactions. That loss motivated our measurement of the coupling efficiency.

Is it a problem that our model is built upon a fuel which is less metastable than a fuel like ATP? We don’t think so. Though our work is partially inspired by biological motors, we are developing a proof-of-principle demonstration, and for two practical reasons we find the demonstrated scenario to be interesting and notable. First, identifying a sweet spot in motor design parameters would be even more challenging if we required a more metastable fuel. To generate motion on a computationally tractable timescale, we would need to construct a particularly effective catalyst. In the present work, we did not need such effective catalytic sites, we only needed sites that would speed up the natural decomposition rate, and that natural rate was already fast enough to see plenty of motion on the simulation timescales. More importantly, we think our demonstration is inspiring for synthetic motors. We show that directionality can be generated even when working with an imperfect fuel.

5. “In a tightly coupled regime one might expect an efficiency of 0.5 (one net cycle for every two catalyzed reactions).” — What does it mean by a tightly coupled regime? and where this expectation of efficiency of 0.5 comes from?

Starting at line 116, we have added a clearer description of the tightness of coupling: “In so-called tightly coupled motors, each reaction event correlates with a configurational change of the motor. For example, when F_1 -ATP synthase generates work from ATP, each catalyzed ATP hydrolysis corresponds almost one-to-one with a 120° rotation of the rotor [65]. Other motors are loosely coupled, with motor motion only weakly correlated with fuel consumption [66].” We have also clarified our reference to a maximal [coupling] efficiency of 0.5, starting at line 310: “We anticipated a maximum coupling of 0.5, corresponding to a tightly coupled cycle with one catalyzed reaction at each catalytic site. Neither motor achieves that limit. Rather, they are loosely coupled, with catalyzed reactions probabilistically gating diffusion and inducing no major conformational changes in the motor itself.” To be even more clear, we are saying that the greatest correlation between fuel decomposition and directed motion would be for each fuel decomposition to advance the shuttling ring halfway in the clockwise direction.

6. The caption of Fig. 3 says that the chemical potential μ'_{FTC} was varied, but then the x-axis is plotted against $\langle N_{\text{FTC}} \rangle$. I understand that Eq. (A9) relates these two quantities. But, if this is the author’s preferred way of presenting their result, the relation between the two quantities should be clarified or Eq. (A9) should be referred to in the text.

This is similar to a comment made by Reviewer 1. We chose concentration over chemical potential because we thought it would give a more intuitive sense of the state of the simulation box. The captions of Fig. 3 have been updated to be more clear on this point, as this reviewer has suggested. Thank you.

7. The subsection “Timescales” aims to show that the chemostat relaxation time is much faster than that of motor dynamics. First, the authors are advised to amend the title to be more specific. Second, please explain why there are the small dips when chemical potential is switched from that of equilibrium to nonequilibrium.

The inclusion of this section, particularly of the figure, led some preprint readers to erroneously assume we were generating current through time-periodic driving rather than in the NESS. For this reason, we decided to move the “Timescales” section (now renamed “Chemostat Timescales”) to the SI. We assume the reviewer is talking about the top plot, (a), and is suggesting that the left boundaries of the shaded regions consistently have a “dip”. Those “dips” occur because the current in the shaded regions is so directed, so any transient downward drift in the white region will necessarily give rise to a local minimum that looks like a dip. Notice, as in the third white region, that the dip does not always appear right at the

boundary between white and gray regions. Rather, it will appear at the most recent time that the white region experienced a transient downward fluctuation due to stochastic fluctuations in the finite number of trajectories.

8. The authors discuss in “the eight state rate model,” indicating that while the R (affinity) is identical for the Motor I and II, the two models differ in terms of their CW/CCW current, efficiency, accuracy . . . , thus pointing that one model is better than the other. Despite the same R , the blocked population is greater for Motor II (Fig.7). Since R is the ratio of all forward rates to all backward rates, there are multiple ways of getting the same R value, which is not at all mysterious. I believe that the authors can demystify the difference of the two Motors by quantifying all the first-order rate constants for transition or all the reaction currents flowing along the edges of network for the Motor I and Motor II completely.

Thank you for this comment. We agree that demystification was a good goal, but we ended up taking this in a different direction than the reviewer suggested. Our initial draft was set up to compare and contrast Motors I and II, but that generated a tremendous amount of redundancy while leaving the mechanism of both motors slightly mysterious. We chose to compare and contrast less while focusing more clearly on the source of the directionality in the new “Clockwise Directionality” section. The reviewer will notice that we still do include discussion of both Motors I and II because we think it is interesting to illustrate that relatively modest tuning of a few pair potentials can significantly alter the CW bias (Figs. 3 and 4).

Reviewers' comments:

Reviewer #1 (Remarks to the Author):

The authors have greatly improved the presentation of their manuscript and have provided convincing answers to the referees' comments. I therefore think that this work is now suitable for publication in Nat. Comm.

The only response I am not fully satisfied with is the one to my point 15. The authors now define the rate k_{sym} as "rotations of the shuttling ring when no blocking groups are present". However, this does not answer my question whether these rotations happen in clockwise or counterclockwise direction. Probably they can happen in either direction, with equal probability as required by symmetry. I find it important to note that there are in fact two pathways for going from state 4 to 6, either in CW or CCW direction (and vice versa for 6→4), as this introduces another fundamental cycle to the network (advancement of the motor without consumption of fuel, which I find just as important as FC4: consumption of fuel without advancement). The distinction of these two pathways is not relevant for the computation of the steady state distribution, but it is relevant for the counting of CW completions of rotations. I see no reason why fuelless completions should not be accounted for (are they?). The symmetric transitions may not affect the current on average, but they do affect the "CW bias". For instance, if k_{sym} is very large, the CW bias will always be close to 0.5. Note that the distinction of CW and CCW rotations would naturally show up if the ring had 3 or more catalytic sites, it is only for the 2-fold symmetry that both directions link between the same two states of the network. (Though I'm not suggesting to augment the discussion to 3-fold symmetry.) The discussion might have been simpler with the discussion of effectively just one catalytic site. I am fine with the authors' decision to stick to two, but the arising subtleties about k_{sym} need to be addressed.

typo l. 341 "shutting ring"

In response to the referee comments, Albaugh and Gingrich have revised the paper extensively. Now, the bias, current, and coupling are clearly defined. The 8-state model and its CW directionality are better explained. The content of the paper is easier to digest than the original version. I find the revised manuscript much improved in terms of the readability. I, however, consider that the paper's impact on our understanding to the principle of molecular motors is only marginal for the publication in Nat. Commun.

1. In the subsection "Clockwise Directionality", the authors delineate the origin of directionality of the motor using the parameter R , which corresponds to the cycle affinity A . The rates, $k_{\text{attach, far}}$ and $k_{\text{attach, close}}$ are the FTC concentration-dependent constant, displaying the Michaelis-Menten type hyperbolic dependences, whereas other rates are independent of it. $R=1$ is for the equilibrium, $R>1$ for CW, and $R<1$ is for CCW... etc.

In my opinion, the authors are still giving a text book type kinetic explanation, without much structural/energetic insight into why such directionality arises from the molecular construct and interaction they explicitly design. It appears to me that in the last paragraph the authors are admitting that they fail in understanding the origin of directionality other than resorting to kinetic explanation based on R .

Given that for the last decades there have been experimental efforts to not only explain the unidirectionality of molecular motors, but also "engineer" the speed, processivity, and even the directionality of the molecular motors (kinesis, myosin, etc), I feel that the authors' effort towards the understanding of the physics underlying this problem is not serious enough.

For examples, the authors should refer to

Nature (1996) 380: 451–453
J Cell Biol (2000) 151 (5): 1093–1100
J. Mol. Biol. (2009) 392, 862–867
PNAS (2007), 104 (3): 772-777
Nat. Commun (2016) 7, 11159

Furthermore, I disagree to the authors comment that the "slope" of $k_{\text{attach, far}}$ and $k_{\text{attach, close}}$ with respect to FTC concentration determines the directionality of motor. According to the expressions of R (Eq.7 and Eq.9), the directionality is determined by the relative magnitude of rate constants at a given FTC concentration, not by their slopes ($dk/d[N_{\text{FTC}}]$?)

2. Instead of actually improving the dynamic behavior of the fuel molecule, the authors only diplomatically addressed my concern as to the quality of fuel molecule.

— Incidentally, the following sentence should be rephrased: "Secondly, the model fuel is not as metastable as ATP" (On Page4, line 284-285). What does it mean by "something is more metastable than others"?

3. My request about resolving the references bias was that the author drastically eliminate many of the references by Astumian (24) and Karpral (14), not merely add 8 additional citations. Still the Ref. [20]-[40] are all by Ray Kapral, which makes the paper appear extremely amateurish. I strongly advise that the authors reduce the number of references by each of these groups to, say, less than 5, leaving only the important ones. The field of molecular motors and molecular simulation is much broader than that implied by the citations of this paper, and its knowledge has been developed and nurtured by the collective effort by the community, not by a few individuals.

"Modeling molecular motors", Jülicher, Ajdari, and Prost, *Rev. Mod. Phys.* **69**, 1269 (1997).

"Molecular motors: A Theorist's Perspective" Kolomeisky, Fisher *Annu. Rev. Phys. Chem.* **58**, 675 (2007)

"*Chemical Biophysics: Quantitative analysis of cellular systems*" Beard and Qian (2008).

“*Mechanics of motor proteins and the cytoskeleton*,” J. Howard (2001).

“The physics of molecular motors” Bustamante, Keller, Oster. *Acc. Chem. Res.* (2001).

“Theoretical perspectives on biological machines”, Mugnai, Hyeon, Hinczewski, Thirumalai, *Rev. Mod. Phys.* **92**, 025001 (2020).

“Nonequilibrium physics in biology” Fang, Kruse, Lu, and Wang, *Rev. Mod. Phys.* **91**, 045004 (2019).

Dear Referees,

Thank you for again providing close readings and thorough reviews. Below you will find point-by-point responses to the reviews.

Regards,

Alex Albaugh and Todd R. Gingrich

Review 1:

The authors have greatly improved the presentation of their manuscript and have provided convincing answers to the referees' comments. I therefore think that this work is now suitable for publication in Nat. Comm.

Thank you for the endorsement. Unfortunately the editor has interpreted your comment about point 15 as reflecting a disagreement with “fundamental aspects of [our] modeling and discussions”. We are sorry our initial response to point 15 wasn't fully illuminating, but we strongly suspect this extended response assuages your concern.

The only response I am not fully satisfied with is the one to my point 15. The authors now define the rate k_{sym} as “rotations of the shuttling ring when no blocking groups are present”. However, this does not answer my question whether these rotations happen in clockwise or counterclockwise direction. Probably they can happen in either direction, with equal probability as required by symmetry. I find it important to note that there are in fact two pathways for going from state 4 to 6, either in CW or CCW direction (and vice versa for $6 \rightarrow 4$), as this introduces another fundamental cycle to the network (advancement of the motor without consumption of fuel, which I find just as important as FC4: consumption of fuel without advancement). The distinction of these two pathways is not relevant for the computation of the steady state distribution, but it is relevant for the counting of CW completions of rotations. I see no reason why fuelless completions should not be accounted for (are they?). The symmetric transitions may not affect the current on average, but they do affect the “CW bias”. For instance, if k_{sym} is very large, the CW bias will always be close to 0.5. Note that the distinction of CW and CCW rotations would naturally show up if the ring had 3 or more catalytic sites, it is only for the 2-fold symmetry that both directions link between the same two states of the network. (Though I'm not suggesting to augment the discussion to 3-fold symmetry.) The discussion might have been simpler with the discussion of effectively just one catalytic site. I am fine with the authors' decision to stick to two, but the arising subtleties about k_{sym} need to be addressed.

Thank you for the comment, which we are very happy to clarify. The reviewer is correct that the symmetric shuttling ring rotations affect the CW bias, but his concerns that this may not be accounted for are unfounded. Recall that the data for CW bias, current, and coupling come directly from the nonequilibrium steady state molecular simulation, not from the eight state model. When collecting the data for Figures 2, 3, and 4, we count the number of cycles (CW and CCW) from the resulting trajectories without any regard for coarse-grained states. These data therefore include fuelless completions and every other type of completion. It is only when calculating rates for the Markov model that we must identify transitions between coarse-grained states. Following the referee's suggestion, we could represent the transition between states 4 and 6 as two pairs of forward and backward edges, both with rates $k_{\text{sym}}/2$. One of those two edges would correspond to clockwise shuttling ring rotations and the other to counterclockwise. Indeed, if we wanted to use the Markov model to predict CW bias then symmetric transitions appear in that calculation in the way the reviewer describes. That calculation was not mentioned in the paper because we instead chose to present CW bias directly measured from NESS simulations, not the bias one would compute from the parameterized Markov model (which, though similar, would include errors associated with the Markov approximation).

One might ask why we included the Markov model if the current, bias, and coupling all came straight from counting the cycles in the simulations. The answer is that the Markov model was employed to understand the directionality of the motor. The distinctions between having a single or double edge between states 4 and 6 do not influence that goal. The reviewer is correct that the double edge would add one more fundamental cycle, but it would be a trivial one that contributes no current by symmetry (it has vanishing affinity). We elected to keep a single forward and reverse edge between states 4 and 6 because the choice doesn't (indeed cannot) impact our logic about the directionality of the motor. Furthermore, our choice is consistent with our treatment of other edges, all of which lump together into a combined rate the contributions from multiple different mechanisms (see, for example, the discussion about Figure 7).

typo l. 341 “shutting ring”

Corrected. Thank you.

Review 2: In response to the referee comments, Albaugh and Gingrich have revised the paper extensively. Now, the bias, current, and coupling are clearly defined. The 8-state model and its CW directionality are better explained. The content of the paper is easier to digest than the original version. I find the revised manuscript much improved in terms of the readability. I, however, consider that the paper's impact on our understanding to the principle of molecular motors is only marginal for the publication in Nat. Commun.

We are pleased that the reviewer has found the paper much improved. Incidentally, we agree that the paper has limited impact on our understanding of the principle of molecular motors, a topic which is very well developed. Importantly, this was not the intention nor the central claim of this work. Rather, this work presents a new class of molecular simulation and a particle-based model to enable direct interrogation of the mapping from pairwise interactions into dynamical function. We think it is notable that this type of nonequilibrium steady state molecular dynamics is absent from the literature, and we feel our development is significant and will be built upon. If the referee strongly disagrees with this belief, we will be very appreciative of demonstrations of prior conflicting work or of clear arguments that our enterprise lacks value.

1. In the subsection “Clockwise Directionality”, the authors delineate the origin of directionality of the motor using the parameter R , which corresponds to the cycle affinity A . The rates, $k_{\text{attach, far}}$ and $k_{\text{attach, close}}$ are the FTC concentration-dependent constant, displaying the Michaelis-Menten type hyperbolic dependences, whereas other rates are independent of it. $R = 1$ is for the equilibrium, $R > 1$ for CW, and $R < 1$ is for CCW...etc. In my opinion, the authors are still giving a text book type kinetic explanation, without much structural/energetic insight into why such directionality arises from the molecular construct and interaction they explicitly design. It appears to me that in the last paragraph the authors are admitting that they fail in understanding the origin of directionality other than resorting to kinetic explanation based on R .

The referee correctly notes that a merit of the sort of model we have built is that one can take the analysis a step further, characterizing how the pairwise interactions sculpt the Michaelis-Menten curves to underpin the “textbook” kinetic descriptions. We agree that this will be possible. We agree that this will be important. We disagree, however, that it belongs in the present manuscript because it turns out to be a particularly nontrivial topic of current work that goes far beyond this manuscript. The pair potentials can be tuned in many different ways, and computing rates from those pair potentials is notoriously difficult. Recall that rate constants vary over many orders of magnitude, and even computing the right order of magnitude is sometimes considered a success. To understand the directionality, however, it is necessary to understand whether R is slightly above or slightly below unity. One can imagine employing approximate rate theories to estimate the rates in terms of energy surfaces and barrier heights, but there is no guarantee that characterizing rates at the level of transition state theory is sufficiently accurate. Rather, the present manuscript presents simulation methodology to sample those rates, showing that the kinetic picture can be systematically *measured* from the described simulations.

Given that for the last decades there have been experimental efforts to not only explain the unidirectionality of molecular motors, but also “engineer” the speed, processivity, and even the directionality of the molecular motors (kinesis, myosin, etc), I feel that the authors’ effort towards the understanding of the physics underlying this problem is not serious enough. For examples, the authors should refer to

Nature (1996) 380: 451–453

J Cell Biol (2000) 151 (5): 1093–1100

J. Mol. Biol. (2009) 392, 862–867

PNAS (2007), 104 (3): 772-777

Nat. Commun (2016) 7, 11159

We thank the reviewer for pointing us to these excellent experimental results. These efforts toward engineering the function of natural molecular motors are very motivating. We certainly agree that these successes have been possible without simulation, but we believe the tool we present will be instructive for designing synthetic supramolecular machines, where the motors cannot be mutated a high-throughput molecular biology fashion. We have significantly rewritten the introduction to try to better place our work in context while better celebrating the demonstrated successes of the reviewer’s molecular motor community.

Furthermore, I disagree to the authors comment that the “slope” of $k_{\text{attach, far}}$ and $k_{\text{attach, close}}$ with respect to FTC concentration determines the directionality of motor. According to the expressions of R (Eq. 7 and Eq. 9), the directionality is determined by the relative magnitude of rate constants at a given FTC concentration, not by their slopes ($dk/d[N_{\text{FTC}}]$)?

Yes, of course Eqs. 7 and 9 show that the important consideration is the ratio of the rate at one concentration to that at zero concentration. We see that it was confusing that we referenced the slope because we were using one other fact: adding more fuel does not change the direction of the motor. In that case the directionality follows from Eq. 9 in the $\langle N_{\text{FTC}} \rangle \rightarrow 0$ limit, hence our discussion of slopes. Referencing these slopes may seem contrived and unnecessary, but recall that $\langle N_{\text{FTC}} \rangle = 0$ is an equilibrium system. By focusing on the slope at $\langle N_{\text{FTC}} = 0 \rangle$, it should be possible to extract the directionality from the response of that equilibrium to a small perturbation (essentially a linear response idea). Because we were not fully developing this idea in the paper, we have removed the mention of the slope by replacing the sentence with “The fuel-dependent attachment rates both increase with FTC concentration, and directionality is determined by which of those rates rises up more rapidly with added N_{FTC} .”

2. Instead of actually improving the dynamic behavior of the fuel molecule, the authors only diplomatically addressed my concern as to the quality of fuel molecule.

We are sorry that our response came across as mere diplomacy. We feel this is an example of the reviewer believing that the paper aims to simulate a biological motor to understand a new fundamental principle. From that perspective, the fuel we work with is “wrong” in that it does not exactly mimic ATP and the work appears useless unless the fuel is “fixed”. We emphasize again that our goal is not the simulation of an ATP-driven motor. We believe physical understanding is aided by the construction of models and by studying the behavior and consequences of those models. Our intention was to introduce one such model with many characteristics of the synthetic molecular motor systems presently being developed. We then aimed to develop and present simulation tools that enable those models to be systematically probed. It’s a goal which is motivated by, but very much tangential, to the goal of explaining/engineering/modeling biological motors.

Incidentally, the following sentence should be rephrased: “Secondly, the model fuel is not as metastable as ATP” (On Page 4, line 284-285). What does it mean by “something is more metastable than others”?

The fuels are metastable in that they are stable species for short times but they decompose over longer times. For something to be more metastable we mean that the timescale for that decomposition is longer. We see this elsewhere in the literature, but we have rephrased the sentence to describe ATP as more “deeply metastable” to evoke the picture of a local minimum which can be more shallow or more deep.

3. My request about resolving the references bias was that the author drastically eliminate many of the references by Astumian (24) and Kapral (14), not merely add 8 additional citations. Still the Ref. [20]-[40] are all by Ray Kapral, which makes the paper appear extremely amateurish. I strongly advise that the authors reduce the number of references by each of these groups to, say, less than 5, leaving only the important ones. The field of molecular motors and molecular simulation is much broader than that implied by the citations of this paper, and its knowledge has been developed and nurtured by the collective effort by the community, not by a few individuals.

“Modeling molecular motors”, Jülicher, Ajdari, and Prost, *Rev. Mod. Phys.* 69, 1269 (1997).

“Molecular motors: A Theorist’s Perspective” Kolomeisky, Fisher *Annu. Rev. Phys. Chem.* 58, 675 (2007).

“Chemical Biophysics: Quantitative analysis of cellular systems” Beard and Qian (2008).

“Mechanics of motor proteins and the cytoskeleton,” J. Howard (2001).

“The physics of molecular motors” Bustamante, Keller, Oster. *Acc. Chem. Res.* (2001).

“Theoretical perspectives on biological machines”, Mugnai, Hyeon, Hinczewski, Thirumalai, *Rev. Mod. Phys.* 92, 025001 (2020).

“Nonequilibrium physics in biology” Fang, Kruse, Lu, and Wang, *Rev. Mod. Phys.* 91, 045004 (2019).

We are sorry that the referee felt that the community was being slighted. Our own interest in the topic was triggered by work on synthetic molecular motors, as well as by work of Astumian and Kapral. We understand the reviewer’s perspective that the manuscript really should be put in an even broader context that includes canonical work on biological motors and their theoretical analysis. Since we had already shared a preprint with the objectionable references, we were initially hesitant to claw back the references we’d already put out there. We have accepted that consideration is secondary and have now followed the referee’s advice to both add and subtract references.

Allow us to highlight, however, that the concern about references specifically relates to the introduction of the paper and the way in which we put our work in context. We honestly appreciate the referee’s perspective, which has helped us improve the paper, but we also feel that introduction is tangential to the heart of our work.

REVIEWERS' COMMENTS

Reviewer #1 (Remarks to the Author):

I cannot see the Editor's comments on my previous report, but, prompted by the authors' response, I would like to stress that my intention of "point 15" was not to point out a fundamental flaw in their model. As the authors rightly conclude, it is a matter of choice which pathways are distinguished and which ones are lumped together in a coarse grained model. I would have made a different choice, but indeed, either choice leads to the same net current. I was aware that the reported CW bias referred to the molecular simulation, my choice would simply have allowed one to calculate the same quantity in the coarse grained model (without making it more complex, since the additional rates are constrained by symmetry). If it only matters that the model produces the right net current, then one could have eliminated the k_{sym} transitions altogether (this could be regarded as lumping the symmetric transitions together with a half-rotation of the ring along with the shuttle, which is not resolved either). But I do not object to the authors' choice, which may be aesthetically more pleasing. The authors should add, upon introduction of the rates k_{sym} , a brief comment about lumping together CW and CCW transitions (just to clarify that the coarse grained model no longer quantifies the CW and CCW fluxes individually). I could imagine that otherwise the reader could be confused, since before the CW bias seemed to have been an important property of the system. If this goes forward towards publication, I trust that the authors will implement this minor change adequately, I do not need to see the manuscript again.

I do not feel entitled to comment on the comments of the other referee. He/she seems to be more rooted in the molecular motors community, where this work may indeed appear as a moderately significant variation on existing motor models. Yet, I stand by my assessment from my first report. From the point of view of stochastic thermodynamics, I think this work is an important step forward. There, models for molecular motors usually start with a discrete model, similar to the coarse grained model discussed here. The present work gives the necessary microscopic underpinning of such models, deriving it from a minimal molecular dynamics model that has been stripped off all the unnecessary details that would be required for a "realistic" biochemical model.

Reviewer #4 (Remarks to the Author):

In my opinion this is a good manuscript that deserves publication in Nature Communications. The starting point is the Brownian ratchet system that is depicted in Figure 1. A small ring (the shuttling ring) is moving around a bigger ring. The preferred positions of the shuttling ring are close to two sites that catalyze a reaction. The product of that reaction can remain bound to the ring for a short but significant time. The attach rate and the cleave rate of that product depend on the nearness of the shuttling ring, i.e., attaching and cleaving are coupled to the position of the shuttling ring. The kinetic scheme in Figure 5 shows how net clockwise motion of the shuttling ring around the bigger ring results. In the end we have a motor as the energy released by the catalyzed reaction is partly

converted into net motion. The authors of the manuscript do an extensive simulation to lead to a realistic accounting of the behavior of this artificial molecular motor. They produce graphs to show how effective the motor is: controllable parameters are varied and depicted are the fraction of clockwise rotations, the net number of clockwise rotations per unit of time, and the net number of clockwise rotations per catalyzed reaction.

Reference 15 is central in the manuscript. The molecular structure that is modeled was created in 2016 by the group of David Leigh. Figure 2 in the manuscript is mindful of Figure 1 in this reference. Reference 15 described qualitatively how this motor should move. This manuscript goes a step further and through huge simulations actually computes the aforementioned graphs to predict how good the structure is as a motor and what the optimal performance parameters are. The currents and clockwise cycles per reaction are, at this stage, not measurable and this manuscript makes predictions that are concrete and may ultimately be verifiable.

The manuscript goes down to the molecular level as far as is computationally feasible. The approach of the reactants near the ring structure is described with Langevin equations. But kinetic rates are still used for the reactions.

An issue here is the novelty and significance. The many pages of computational detail are a demanding read and there is never a big "gee whiz" kind of surprise in the manuscript. The insights about Brownian ratchets were developed in the 1990s. This manuscript adds a lot of concrete chemical and computational detail to these insights. The manuscript also adds quantitative predictions to Reference 15. As Reference 15 appeared in Nature, it is only sensible that significant further development (predictions of generated motor current!) also appears in Nature.

I have little criticism and almost no suggestions for revision. The manuscript is very well proofread. The only typo that I found is in the legend of Figure 5 where a "the" is missing in front of "shuttling." In the graph on the right side of Figure 7, the curves are bluish green and greenish blue. More distinct colors is probably a good idea. The "Thermodynamics" section is the only more theoretical tangent in the manuscript. I appreciate the point that the authors make towards the end of that section; a point they illustrate with Figure 7. However, the beginning of the section is somewhat obfuscating. In different contexts the terms "detailed balance" and "microscopic reversibility" sometimes mean different things. But in the context of Figure 5 it should be clear. If there is net clockwise cycling, i.e. NESS, then detailed balance is violated. The violation of detailed balance implies violation of microscopic reversibility. Equilibrium and microscopic reversibility imply each other. The authors should be either more clear or cut the wordiness. Detailed balance and microscopic reversibility do not really have to be involved at all here.

Dear Referees,

We once again thank you for your time, consideration, and insightful comments. We believe that this manuscript has been substantially improved by your contributions.

Regards,

Alex Albaugh and Todd R. Gingrich

Review 1:

I cannot see the Editor's comments on my previous report, but, prompted by the authors' response, I would like to stress that my intention of "point 15" was not to point out a fundamental flaw in their model. As the authors rightly conclude, it is a matter of choice which pathways are distinguished and which ones are lumped together in a coarse grained model. I would have made a different choice, but indeed, either choice leads to the same net current. I was aware that the reported CW bias referred to the molecular simulation, my choice would simply have allowed one to calculate the same quantity in the coarse grained model (without making it more complex, since the additional rates are constrained by symmetry). If it only matters that the model produces the right net current, then one could have eliminated the k_{sym} transitions altogether (this could be regarded as lumping the symmetric transitions together with a half-rotation of the ring along with the shuttle, which is not resolved either). But I do not object to the authors' choice, which may be aesthetically more pleasing. The authors should add, upon introduction of the rates k_{sym} , a brief comment about lumping together CW and CCW transitions (just to clarify that the coarse grained model no longer quantifies the CW and CCW fluxes individually). I could imagine that otherwise the reader could be confused, since before the CW bias seemed to have been an important property of the system. If this goes forward towards publication, I trust that the authors will implement this minor change adequately, I do not need to see the manuscript again.

The reviewer's comments on the ambiguity of k_{sym} are correct. We have taken their suggestion and added the following sentence after k_{sym} is introduced in the text: "The rates k_{CW} and k_{CCW} unambiguously imply a direction of shuttling ring motion, while k_{sym} results in an even split between clockwise and counterclockwise."

I do not feel entitled to comment on the comments of the other referee. He/she seems to be more rooted in the molecular motors community, where this work may indeed appear as a moderately significant variation on existing motor models. Yet, I stand by my assessment from my first report. From the point of view of stochastic thermodynamics, I think this work is an important step forward. There, models for molecular motors usually start with a discrete model, similar to the coarse grained model discussed here. The present work gives the necessary microscopic underpinning of such models, deriving it from a minimal molecular dynamics model that has been stripped off all the unnecessary details that would be required for a "realistic" biochemical model.

Review 4:

In my opinion this is a good manuscript that deserves publication in Nature Communications. The starting point is the Brownian ratchet system that is depicted in Figure 1. A small ring (the shuttling ring) is moving around a bigger ring. The preferred positions of the shuttling ring are close to two sites that catalyze a reaction. The product of that reaction can remain bound to the ring for a short but significant time. The attach rate and the cleave rate of that product depend on the nearness of the shuttling ring, i.e., attaching and cleaving are coupled to the position of the shuttling ring. The kinetic scheme in Figure 5 shows how net clockwise motion of the shuttling ring around the bigger ring results. In the end we have a motor as the energy released by the catalyzed reaction is partly converted into net motion. The authors of the manuscript do an extensive simulation to lead to a realistic accounting of the behavior of this artificial molecular motor. They produce graphs to show how effective the motor is: controllable parameters are varied and depicted are the fraction of clockwise rotations, the net number of clockwise rotations per unit of time, and the net number of clockwise rotations per catalyzed

reaction.

Reference 15 is central in the manuscript. The molecular structure that is modeled was created in 2016 by the group of David Leigh. Figure 2 in the manuscript is mindful of Figure 1 in this reference. Reference 15 described qualitatively how this motor should move. This manuscript goes a step further and through huge simulations actually computes the aforementioned graphs to predict how good the structure is as a motor and what the optimal performance parameters are. The currents and clockwise cycles per reaction are, at this stage, not measurable and this manuscript makes predictions that are concrete and may ultimately be verifiable.

The manuscript goes down to the molecular level as far as is computationally feasible. The approach of the reactants near the ring structure is described with Langevin equations. But kinetic rates are still used for the reactions.

An issue here is the novelty and significance. The many pages of computational detail are a demanding read and there is never a big “gee whiz” kind of surprise in the manuscript. The insights about Brownian ratchets were developed in the 1990s. This manuscript adds a lot of concrete chemical and computational detail to these insights. The manuscript also adds quantitative predictions to Reference 15. As Reference 15 appeared in Nature, it is only sensible that significant further development (predictions of generated motor current!) also appears in Nature.

I have little criticism and almost no suggestions for revision. The manuscript is very well proofread. The only typo that I found is in the legend of Figure 5 where a “the” is missing in front of “shuttling.”

This typo has been corrected.

In the graph on the right side of Figure 7, the curves are bluish green and greenish blue. More distinct colors is probably a good idea.

We appreciate the suggestion and have changed the color scheme in Figs. 5, 6, and 7 to be more distinguishable.

The “Thermodynamics” section is the only more theoretical tangent in the manuscript. I appreciate the point that the authors make towards the end of that section; a point they illustrate with Figure 7. However, the beginning of the section is somewhat obfuscating. In different contexts the terms “detailed balance” and “microscopic reversibility” sometimes mean different things. But in the context of Figure 5 it should be clear. If there is net clockwise cycling, i.e. NESS, then detailed balance is violated. The violation of detailed balance implies violation of microscopic reversibility. Equilibrium and microscopic reversibility imply each other. The authors should be either more clear or cut the wordiness. Detailed balance and microscopic reversibility do not really have to be involved at all here.

We agree that the use of the terms “detailed balance” and “microscopic reversibility” can lead to misunderstanding and confusion among different communities. At the referee’s suggestion, we have reworked the section to make it both clearer and less verbose.